# Annelid adult cell type diversity and their pluripotent cellular origins

Patricia Álvarez-Campos [1,2,8] ✉, Helena García-Castro [1,7,8], Elena Emili[1], Alberto Pérez-Posada[1,7], Irene del Olmo [2], Sophie Peron [1,7], David A. Salamanca-Díaz [1,7], Vincent Mason[1], Bria Metzger[3,4], Alexandra E. Bely[5], Nathan J. Kenny [1,6], B. Duygu Özpolat [3,4] ✉ & Jordi Solana [1,7] ✉

Many annelids can regenerate missing body parts or reproduce asexually, generating all cell types in adult stages. However, the putative adult stem cell populations involved in these processes, and the diversity of cell types generated by them, are still unknown. To address this, we recover 75,218 single cell transcriptomes of the highly regenerative and asexually-reproducing annelid *Pristina leidyi*. Our results uncover a rich cell type diversity including annelid specific types as well as novel types. Moreover, we characterise transcription factors and gene networks that are expressed specifically in these populations. Finally, we uncover a broadly abundant cluster of putative stem cells with a pluripotent signature. This population expresses well-known stem cell markers such as *vasa*, *piwi* and *nanos* homologues, but also shows heterogeneous expression of differentiated cell markers and their transcription factors. We find conserved expression of pluripotency regulators, including multiple chromatin remodelling and epigenetic factors, in *piwi*+ cells. Finally, lineage reconstruction analyses reveal computational differentiation trajectories from *piwi*+ cells to diverse adult types. Our data reveal the cell type diversity of adult annelids by single cell transcriptomics and suggest that a *piwi*+ cell population with a pluripotent stem cell signature is associated with adult cell type differentiation.

Most annelid species can regenerate at least some body parts and continuously add new body segments from a posterior growth zone throughout their lives. Many are also capable of asexual reproduction by fragmentation or fission. Therefore, many annelids can generate and regenerate all adult cell types from pieces of the adult body[1,2]. However, the cellular and molecular mechanisms of adult cell differentiation are still poorly understood. Cell proliferation is spatially highly localised during adult forms of development in annelids, with proliferation being concentrated in the tip of the tail during segment addition, in mid-body zones during fission, and at the wound site during regeneration. Within these proliferative zones, large numbers of cells that express conserved stem cell markers have been detected,

[1]Department of Biological and Medical Sciences, Oxford Brookes University, Oxford, UK. [2]Centro de Investigación en Biodiversidad y Cambio Global (CIBC-UAM) & Departamento de Biología (Zoología), Facultad de Ciencias, Universidad Autónoma de Madrid, Madrid, Spain. [3]Eugene Bell Center for Regenerative Biology and Tissue Engineering, Marine Biological Laboratory, 7 MBL Street, Woods Hole, MA 05432, USA. [4]Department of Biology, Washington University in St. Louis. 1 Brookings Dr. Saint Louis, Saint Louis, MO 63130, USA. [5]Department of Biology, University of Maryland, College Park, MD 20742, USA. [6]Department of Biochemistry, University of Otago, P.O. Box 56 Dunedin, Aotearoa, New Zealand. [7]Present address: Living Systems Institute, University of Exeter, Exeter, UK. [8]These authors contributed equally: Patricia Álvarez-Campos, Helena García-Castro. ✉e-mail: patricia.alvarez@uam.es; bdozpolat@wustl.edu; jsolana@brookes.ac.uk

suggesting a role for stem cells in these processes. For example, during posterior growth, high concentrations of cells expressing stem cell markers *piwi* and *vasa*, among others, are found in the segment addition zone[3]. During fission, cells expressing *piwi*, *vasa*, and *PL10* are highly concentrated in early to mid-stage fission zones of species of *Pristina*[4–6]. During regeneration, expression of several pluripotent cell markers is initiated at the wound site seemingly de novo in species of *Capitella* and *Pristina*, suggestive of a de-differentiation process[5–11], and in a species of *Enchytraeus*, there is also evidence of cells expressing *piwi* migrating toward wound sites to participate in regeneration[12–14]. To understand how annelids continuously produce new differentiated cells as juveniles and adults during posterior growth, asexual fission and regeneration, it is key to elucidate how many cell types are present in adult annelids, and to reconstruct their differentiation trajectories.

Tracing developmental cell lineages is remarkably difficult in adult animal models without well-developed transgenesis. Single cell transcriptomics (scRNA-seq) has emerged as a powerful tool to study the cellular composition – the *cell type atlas* – of multicellular organisms[15]. But, importantly, scRNA-seq has also fuelled the development of lineage reconstruction algorithms[16]. These algorithms order cells in their differentiation trajectory, revealing the genetic changes that underlie the transition from stem cell to differentiated cell types. Making use of this powerful approach, differentiation trajectories have been reconstructed in adult cell type differentiation models such as planarians[17,18], acoels[19,20], cnidarians[21], sponges[22], and amphibians[23,24].

Cell-type atlases of embryonic, larval and adult annelids have previously been generated[25–28]. However, despite the multiplication of single-cell atlas studies in diverse metazoan species, annelid adult cell types and their differentiation trajectories are still uncharacterised. *Pristina leidyi* (hereafter referred to as *Pristina*) is a convenient laboratory model annelid to address these questions[29,30]. It grows very rapidly in culture conditions by asexual reproduction, using a mechanism called paratomic fission, in which the worm starts forming and differentiating new head and tail segments from within a single body segment, producing a chain of worms[30]. Eventually, these clones separate and become distinct individuals. Thus, these worms are constantly generating all body parts and therefore all adult cell types. Three different zones of intense proliferation have been described in adult *Pristina* worms by S-phase cell EdU/BrdU labelling, located in the anterior end, the posterior end and the fission zones[30,31]. These areas also contain large numbers of *piwi*+, *nanos*+ and *vasa*+ cells[5,6]. This molecular signature has been associated with the stem cells of very diverse invertebrates[32–35]. The transcriptome of these cells has been profiled in some organisms, giving insight into their expression patterns and their heterogeneity, which reflects their developmental potency. For instance, the stem cell pool in planarians contains stem cells that coexpress *piwi* with transcription factors characteristic of differentiated cell types[36–40]. However, in annelids, the transcriptional profiles of *piwi*+ cells and their differentiated counterparts are still unknown.

Here we used scRNA-seq to profile the adult cell type atlas of *Pristina* and reconstruct its differentiation trajectories. We characterised all major adult cell types and uncovered an abundant *piwi*+ cell cluster with a clear stem cell signature. We reconstructed *piwi*+ cell differentiation trajectories to diverse cell types, a signature of pluripotency. We also showed that this population is heterogeneous, indicating the presence of committed stem cells. Finally, we characterised the molecular signature of annelid *piwi*+ cells at the transcriptional level, revealing a transcriptional program composed of RNA binding proteins, cell cycle control, DNA repair mechanisms, and chromatin regulators. Our data show that adult cell type differentiation in *Pristina* is underlied by a *piwi*+ cell population with a pluripotent stem cell signature.

## Results

### A cell-type atlas of the annelid *Pristina leidyi*

We first obtained a new transcriptome from adult *Pristina* individuals (mixed stages, mRNA) using Iso-Seq. Of the 29,807 transcripts, we annotated 18,551 transcripts using eggNOG[41] and 19,582 transcripts using Diamond BLAST[42,43] (18,114 transcripts overlap, Supplementary Data 1, Supplementary Note 1). We then used ACME[44] to obtain cell dissociations of adult mixed populations of *Pristina* containing intact organisms in all fissioning stages (Fig. 1A) and performed three independent single-cell transcriptomic experiments using SPLiT-seq[45] (Fig. 1A) with 4 rounds of combinatorial barcoding. We obtained a total of 80,387 cell profiles and used Scrublet[46] and Solo[47] to eliminate 4966 cells (6.1%) as potential doublets (Supplementary Fig. 1, Supplementary Note 1). We explored the preprocessing parameter space with the remaining 75,421 cells (Supplementary Data 2, Supplementary Fig. 2, Supplementary Note 1) and then clustered the dataset with the Leiden algorithm at resolution 1.5. This allowed us to robustly identify 60 cell clusters (Fig. 1B, Supplementary Fig. 2C-D, Supplementary Fig. 3A) that are reproducible across parameter conditions (Supplementary Data 2), and have highly specific markers (Fig. 1C, Supplementary Fig. 2, Supplementary Data 5). We left some small clusters unannotated as further potential doublets (46, 47, 48, 50, 51, 52, 53, 54, 56, 57, 58, ranging from 174 to 41 cells, 0.2% and 0.05% of the dataset, respectively, Supplementary Note 1).

We then performed PAGA[48] using only annotated clusters (Fig. 1D, Supplementary Note 1) to reconstruct differentiation trajectories. PAGA estimates connectivity within clusters that can be interpreted as computationally inferred lineage relationships. This lineage reconstruction allowed us to classify the broad cell types (Fig. 1E). We also performed a co-occurrence analysis of cell type clusters[49], using the gene expression data of highly variable genes, summed at the cell cluster level. This analysis broadly confirmed our cluster groups (Supplementary Fig. 5). We annotated individual cell types and group identities by considering their gene markers within the context of the published annelid literature, the lineage reconstruction and the in situ Hybridisation Chain Reaction (HCR) characterisation (Fig. 1E, Supplementary Note 2).

### In situ *HCR* validates epidermal, muscular and neuronal identities and reveals high antero-posterior regionalisation of the gut in *Pristina leidyi*

We developed a multiplexed in situ HCR protocol for *Pristina* and validated most cluster identities using specific cluster markers (Supplementary Data 6, Supplementary Fig. 6). First, we characterised major cell types such as epidermis, neurons, and muscle (Fig. 2A). We characterised the epidermis based on the expression of PrileiEVm008309t1. This marker was found all across the outer body wall and along the entire length of the worm's body (Fig. 2B). Neural populations were defined based on the expression of *synaptotagmin* (PrileiEVm012030t1) and validated by in situ expression of PrileiEVm000558t1, a broad neuronal marker. We found staining anteriorly in the head and in ventral clusters of neurons across the body, reminiscent of previously published immunostainings for neurons[30,50] (Fig. 2C). Finally, we characterised muscle clusters based on their high expression of muscle markers (e.g. *myosin*, *tropomyosin*, *troponin*). The in situ hybridisation of one of these markers, the myosin heavy chain homologue gene PrileiEVm000300t1, revealed longitudinal muscle fibres extending along the surface of the animal (Fig. 2D).

We identified 10 gut and gut-associated cell clusters (Fig. 3A), and visualised the localisation of their markers using in situ HCR (Fig. 3B–J). These analyses revealed that *Pristina* has a complex gut organisation with specific molecular regions and cell types along the entire antero-posterior axis. Some of these regions were restricted to as few as 2 segments, such as the crop region (cluster 31) which always occurred in segments 5–7 (Fig. 3B–E; Supplementary Fig. 7). Some gut markers exhibited consistent and sharp borders. In all samples analysed, the

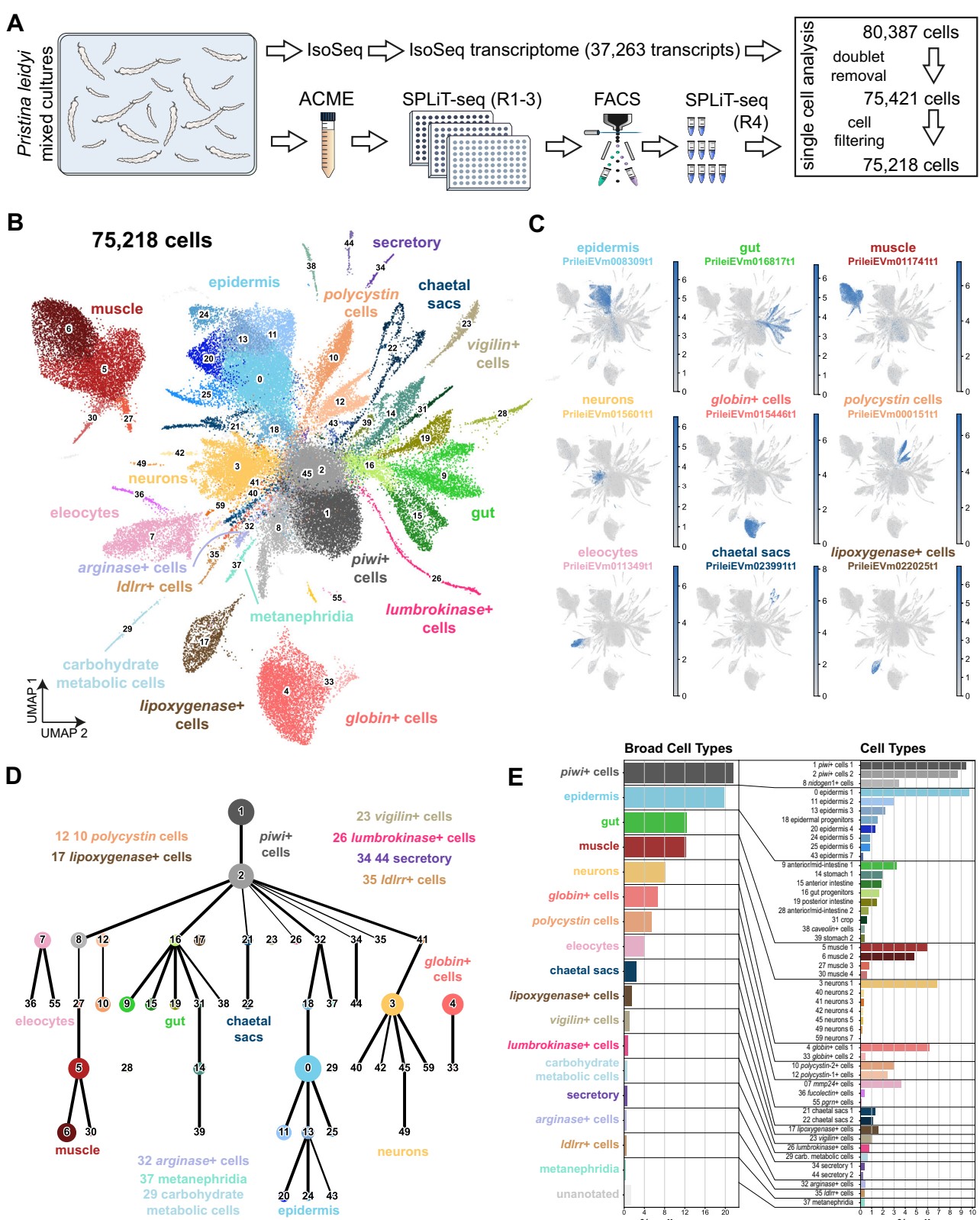

**Fig. 1 | Single-cell atlas of adult *Pristina leidyi* annelids. A** Experimental workflow. Cartoon adapted from reference. 44. **B** UMAP visualisation of the 75,218-cell *Pristina leidyi* single-cell transcriptomic atlas with clusters coloured according to their cell identity. **C** Expression plots of markers of the major broad types, including epidermis, gut, muscle, neurons, *globin+* cells, *polycystin* cells, eleocytes, chaetal sacs and *lipoxygenase+* cells. **D** Lineage reconstruction abstracted graph (PAGA)

showing the most probable path connecting the clusters. Each node corresponds to a cell cluster identified with the leiden algorithm. The size of nodes is proportional to the amount of cells in the cluster, and the thickness of the edges is proportional to the connectivity probabilities. Nodes are coloured according to their cell identity. **E** Cell cluster percentages at the broad cell type and the cell type levels.

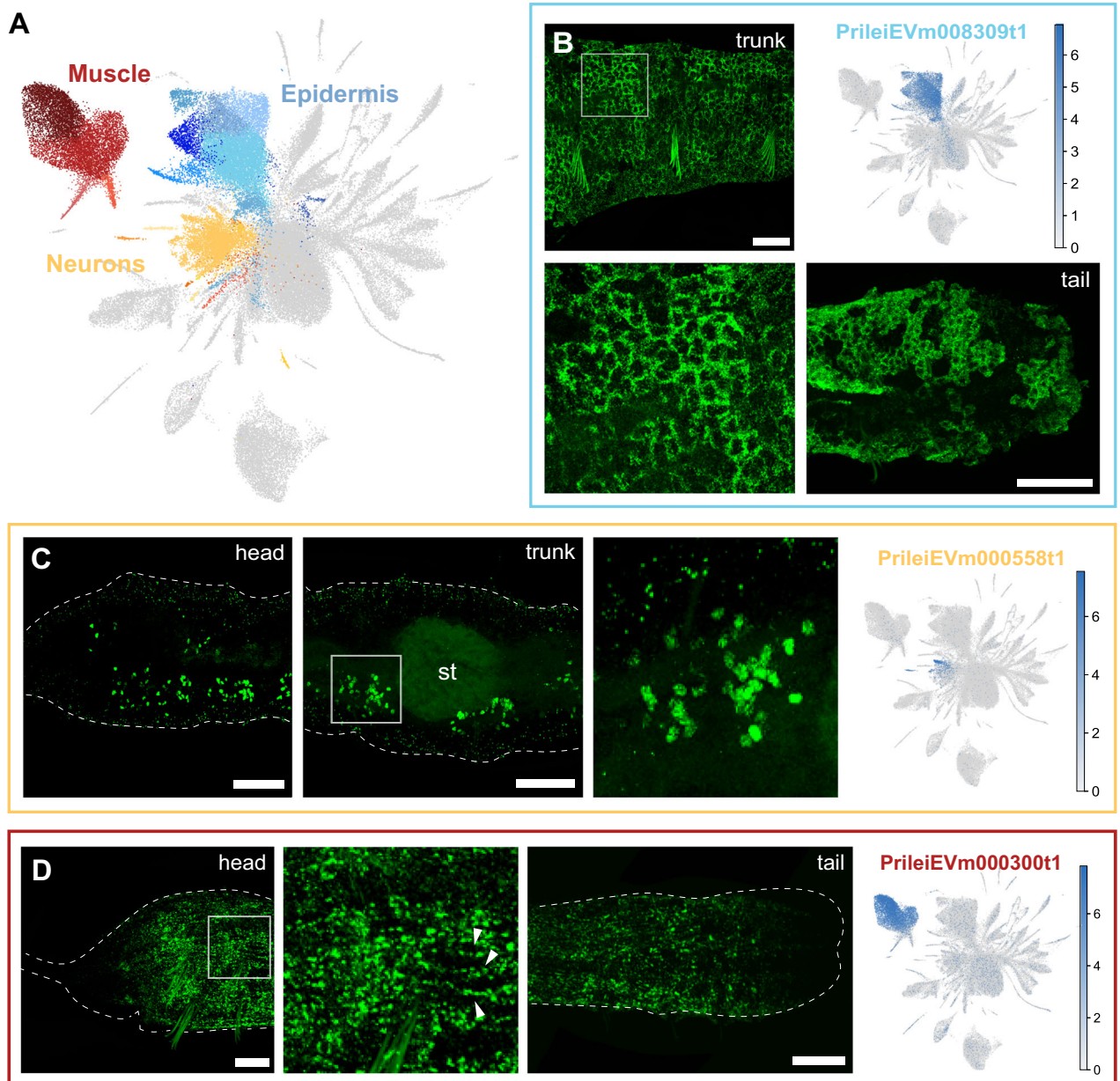

**Fig. 2 | Epidermal, muscle and neuronal clusters in *Pristina leidyi*. A** – UMAP visualisation highlighting Epidermis (blue), Muscle (red), and Neuron (yellow) clusters. **B** – In situ HCRs and expression plot of epidermis marker PrileiEVm008309t1, showing extensive signal in the epidermal cells across the body. The bottom left panel is a close-up of the top left panel. **C** – In situ HCR and expression plot of the neuronal marker PrileiEVm000558t1, showing groups of neuronal cell bodies across the worm's body. The right microscopy panel is a close-up from the middle microscopy panel. **D** – In situ HCR and expression plot of the muscle marker PrileiEVm000300t1, showing expression along the worm. The middle microscopy panel is a close-up from the left panel, evidencing muscle fibres (arrowheads). Scale bars are 50 μm unless otherwise specified. All expression patterns displayed in the figure were observed in, at least, 3 different individuals.

crop and stomach (clusters 14 and 39) always had a sharp border with cells at this boundary expressing either the crop marker or the stomach marker, but never both (Fig. 3E). Similarly, the most posterior gut marker (PrileiEVm021761t1) was always expressed up until the anus, largely coincident with a region with long cilia in the posterior intestine[51]. In contrast, some markers were expressed in broadly the same regions of the gut, but their cellular expression did not overlap (Fig. 3H, I), indicating the presence of distinct cell types in those regions. Among them, we found a cell cluster with high expression of lumbrokinase enzymes (cluster 26, Fig. 3E), identifying the cell type that produces this previously described fibrinolytic enzyme[52]. The expression of intestine markers along the anterior-posterior axis tended to be proportional to the worm's overall length, suggesting that

these gut regions expand proportionally as the worms grow longer (Fig. 3B, Supplementary Fig. 7). These results show that our single cell data resolve the complex gut organisation of *Pristina*, with distinct molecular regions along the anterior-posterior axis and several regionally specific cell types.

## Single cell transcriptomics reveals a wealth of annelid cell types and novel cell types

We then aimed to characterise the remaining set of clusters (Fig. 4A). We identified previously described annelid cell types as well as novel cell types. For instance, we identified a population of *ldlrr*+ cells (cluster 35), which are distributed throughout the animal (Fig. 4B) and have a morphology with numerous extensions (Fig. 4B, inset),

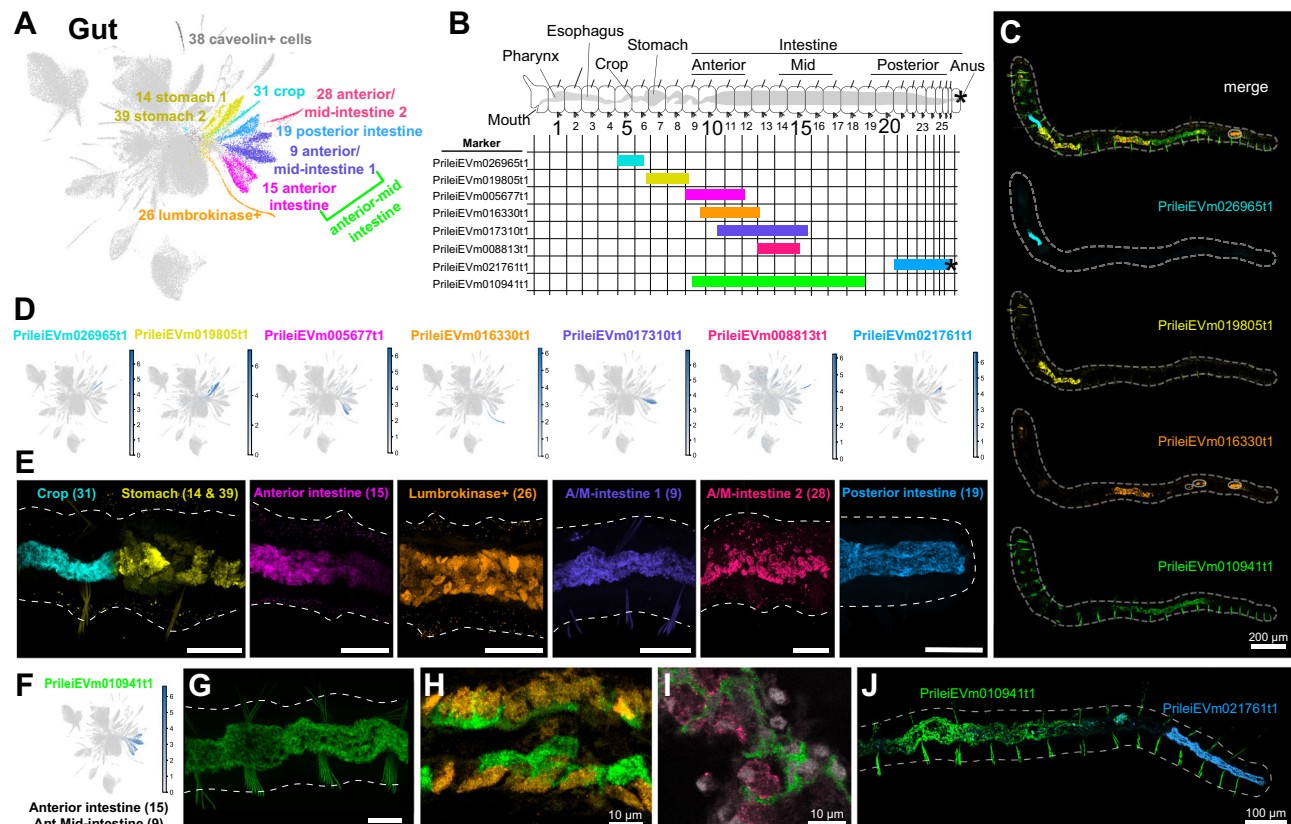

**Fig. 3 | Gut organisation of *Pristina leidyi*. A** UMAP visualisation highlighting gut and associated clusters. The colour code matches the colours in the microscopy images. **B** General distribution of marker expression representing each gut region along the worm (see Supplementary Fig. 7). Cartoon adapted with permission from references. 5 and 69. **C** Example in situ HCR showing 4 different markers simultaneously, but in distinct regions of the gut (blue, yellow, orange, green). Dashed line indicates the outline of the worm. Circles indicate background signal in the gut. **D** Expression plots of diverse gut cluster markers. Gene colour code matches the colours of the clusters. **E** In situ HCR expression of diverse gut cluster markers. All images are lateral views. Note the strict border between the crop and stomach, where there is no co-expression of the markers. **F** Expression plot of anterior and mid intestine marker PrileiEVm010941t1, with expression in cell clusters 9 and 15. **G** In situ HCR expression of PrileiEVm010941t1. **H** In situ HCR expression of PrileiEVm010941t1 (green) and lumbrokinase+ cell marker PrileiEVm016330t1 (orange), showing non overlapping expression in the same gut region. **I** In situ HCR expression of PrileiEVm010941t1 (green) and anterior/mid-intestine marker PrileiEVm008813t1 (pink), showing non overlapping expression in the same gut region. **J** In situ HCR expression of PrileiEVm010941t1 and posterior intestine marker PrileiEVm021761t1, showing non overlapping expression in distinct gut regions. In all panels, anterior is left and dorsal is up (unless otherwise noted). Tail in (**B**) is ventrolateral. Scale bars are 50 μm unless otherwise specified. All expression patterns displayed in the figure were observed in, at least, three different individuals.

reminiscent of astrocytes[53]. Furthermore *ldlrr*+ cells express PrileiEVm006872t1, a homologue of the intermediate filament *gliarin*[54]. We also identified a population of cells (cluster 29) located in the posterior gut, up to 3–4 segments before the tail end. These cells express Krebs cycle and mitochondrial enzymes and we therefore refer to them as carbohydrate metabolic cells (Fig. 4C). We did not find previous descriptions of these populations in the annelid literature and therefore considered these novel cell types.

We also found clusters that likely represent cell types previously described in annelids at the morphological or molecular level. For instance, we found that clusters 7 and 36 express the marker vitellogenin and likely correspond to eleocytes, a type of coelomocyte with a nutritive role and involved in annelid yolk synthesis[55]. In *Pristina*, eleocytes were present in the dorsal side and around the gut across the whole body (Fig. 4D). We also found a prominent cell population (clusters 4 and 33) that expressed several extracellular globins (Supplementary Data 3–4). Although in the annelid *Platynereis dumerilii* such globins are expressed in transverse trunk vessels and parapodial vessels ii[56], we found that *globin*+ cells in *Pristina* occupy large areas in the vicinity of the gut (Fig. 4E). Then, we identified a cluster (23) marked by the expression of *vigilin*, an RNA-binding protein important for chromosome stability and cell ploidy[57]. In *Drosophila* and humans, the *vigilin* homologue, DDP1, interacts with mRNAs localised in the

endoplasmic reticulum[58,59]. *Pristina vigilin*+ cells are located in three large bulbs in the anterior segments of the worm (Fig. 4F). Based on their location and morphology, these likely correspond to pharyngeal glands, which have been described in many oligochaetes, including species of *Pristina*[60,61]. Interestingly, this cluster showed a higher number of RNA UMI counts per cell (Supplementary Fig. 2D). We wondered if this was a technical artefact or a biological observation instead, with *vigilin*+ cells being larger cells. We quantified the cell nuclei area of *vigilin*+ cells and determined that their size is significantly larger than that of other cells (Fig. 4E). This large size could be a product of polyploidisation, but could also be a consequence of increased transcriptional activity or a higher amount of open chromatin[62]. Furthermore, we found a transcript encoding a mucin gene in the marker list. Together, our results characterise this cell type as pharyngeal glands from morphological, cytological and transcriptional data, but this interesting finding would require further work in order to suggest their potential function and diversification within Annelida.

We then examined two prominent and abundant (3.1% and 2.4%) clusters marked by *polycystin* genes, a family of genes associated with cilia[63]. We found that *polycystin-2*+ cells (cluster 10) were segmentally repeated in the body wall of the worm (Fig. 4G), likely corresponding to sensory cells equipped with ciliary tufts[64,65]. In contrast, *polycystin-1*+ cells (cluster 12) were enriched in the head segments (Fig. 4G). We

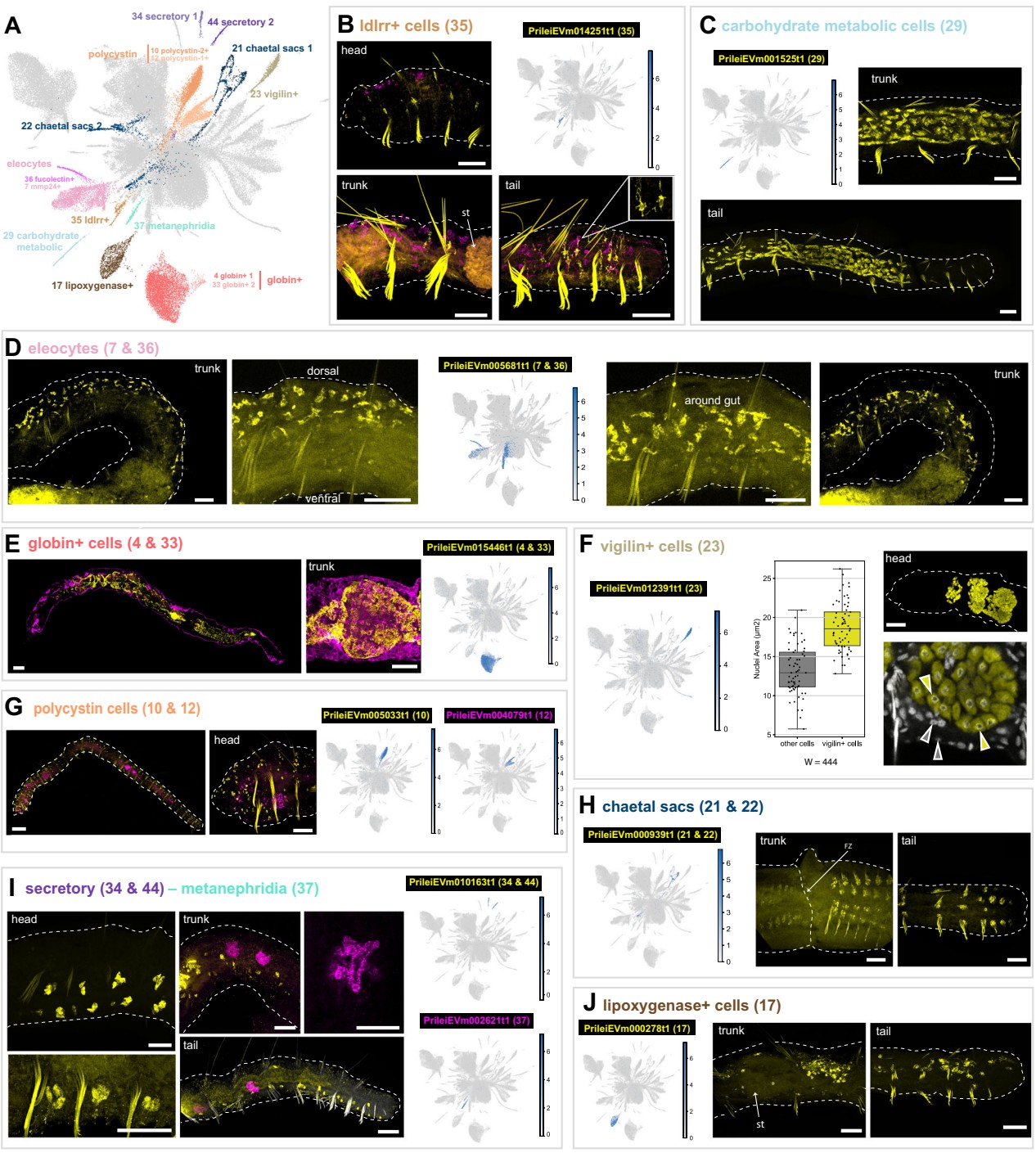

also found that clusters 21 and 22 corresponded to the chaetal sacs (Fig. 4H), which were marked by the expression of a transcript encoding a *chitin synthase protein* (PrileiEVm000573t1). Clusters 34 and 44 corresponded to segmentally repeated cells all along the body of the animal, with a likely secretory function (Fig. 4I), based on the expression of a conotoxin protein (PrileiEVm010163t1). Cluster 37 corresponded to the metanephridia with a clear tubular structure (Fig. 4I). Finally, *lipoxygenase*+ cells (cluster 17) were characterised by the expression of numerous lipoxygenase enzymes (Supplementary Data 3–4). These fatty acid-peroxidising enzymes are involved in a range of immune, signalling and metabolic functions[66]. *Lipoxygenase*+ cells are large cells distributed throughout the AP axis of the animal (Fig. 4J), and could correspond to the previously described chloragocytes[67].

Altogether, these observations identified several annelid cell types such as the eleocytes, the *globin*+ cells, the *vigilin*+ cells, the *polycystin* cells, the chaetal sacs, the metanephridia and the *lipoxygenase*+ cells, but also revealed previously unknown cell types such as the *ldlrr*+ cells and the carbohydrate metabolic cells, with function and homologies that are yet to be explored. Thus, our single cell dataset reveals new biological insights into blood-related cell types and metabolic cell types among others, opening up numerous research avenues for annelid researchers and for the investigation of the evolution of cell types.

## The transcriptional landscape of annelid adult cell differentiation

We then investigated the specific gene expression patterns of each *Pristina* cell type. Given the low UMI and gene counts of our

**Fig. 4 | Annelid specific and novel cell types. A** UMAP visualisation highlighting annelid specific and novel cell types. **B** In situ HCR and expression plot of the *ldlrr*+ cell marker (cluster 35) PrileiEVm014251t1, showing signal throughout the whole animal body. Detail of the extensions of *ldlrr*+ cells is shown in the inset of the tail picture. Magenta counterstaining corresponds to eleocytes and *nidogen*+ cell marker (clusters 7 and 36) PrileiEVm005681t1. **C** In situ HCR and expression plot of carbohydrate metabolic cells marker (cluster 29) PrileiEVm001525t1, showing extensive signal in the posterior end of the animal. **D** In situ HCR and expression plot of eleocyte cell marker (clusters 7 and 36) PrileiEVm005681t1, showing expression in the dorsal area and around the animal's gut. **E** In situ HCR and expression plot of *globin*+ cell marker (clusters 4 and 33) PrileiEVm015446t1, showing expression around the animal's gut. Magenta staining corresponds to epidermal marker PrileIEVm008309t1. **F** In situ HCR and expression plot of *vigilin*+ cell marker (cluster 23) PrileiEVm012391t1, in the anterior part of the animal. Barplot shows nuclei area quantification on a sample size of $n = 130$, 65 *vigilin*- nuclei (grey) and 65 *vigilin*+ nuclei (yellow), examined over 3–5 focal planes in three different animals. Barplot squares represent the median line, and lower and upper quartiles. Whiskers represent sample minimum and maximum values. Median is

12.9 μm² for *vigilin*- cells and 18.6 μm² for *vigilin*+ cells. A statistical Wilcoxon test ($W = 444$, *p*-value = 8.009e−15) indicates significant differences between the two groups. **G** In situ HCR and expression plots of *polycystin* cell markers (clusters 10 and 12) PrileiEVm005033t1 and PrileiEVm004079t1, showing the expression of *polycystin-2*+ cells (yellow) segmentally repeated throughout the body wall of the animal and *polycystin-1*+ (magenta) only in the anterior region. **H** In situ HCR and expression plots of chaetal sacs markers (cluster 21 and 22) PrileiEVm000939t1, showing the expression in the fission zone (FZ) and in the tail of the animal. **I** In situ HCR and expression plots of secretory (cluster 34 and 44) and metanephridia (clusters 37) markers PrileiEVm010163t1 and PrileiEVm002621t1, respectively. Secretory cells are segmentally repeated, mostly ventrally, all along the whole body of the animal. Metanephridia cells show expression in some specific segments of the midbody and posterior regions. **J** In situ HCR and expression plot of *lipox-ygenase*+ cell marker (cluster 17) PrileiEVm000278t1, showing cell expression in the trunk, around the stomach (st), and posterior parts of the animals. In most panels, anterior is left and dorsal is up, except trunk in (**H**), which is ventrolateral. All scale bars are 50 μm. All expression patterns displayed in the figure were observed in, at least, three different individuals.

combinatorial single cell dataset, we used a pseudobulk approach, aggregating raw reads coming from all cells in each cluster. This allowed us to quantify a mean of 11,117 genes per cluster (Supplementary Fig. 8A). We then used Weighted Gene Coexpression Network Analysis (WGCNA)[68] to identify genes with correlated expression patterns. We identified 10,796 genes distributed over forty modules of specific gene expression, broadly corresponding to most cell clusters identified (Fig. 5A, Supplementary Data 7). We used Gene Ontology analysis to extract biologically relevant terms for each cell type (Fig. 5B, Supplementary Data 8). For instance, the module *cilia* corresponded to genes expressed in several cell types but enriched in cilia-related GO terms (Fig. 5A, Supplementary Data 8). To assess the potential regulatory layer underlying this transcriptional landscape, we focused our attention on Transcription Factors (TFs). We annotated 958 *Pristina* TFs (see Methods, Supplementary Data 9, Supplementary Fig. 8B–E) and identified cell type-specific expression of dozens of them (Fig. 5C), including well-known markers or regulators of several cell types, such as a *pou-6f* gene in neurons and a *myoD* gene in muscle (Fig. 5D). This included rich regulatory detail, for instance in the gut, with *hnf4* and *nkx-2-1* TFs broadly expressed in gut clusters, but excluded from *lumbrokinase*+ cells, and a *gata-4* TF with similar expression, but including the *lumbrokinase*+ cells (Fig. 5D, asterisk). This analysis allowed us to obtain insight for the first time into our annelid specific and novel cell types, identifying TFs specific to eleocytes, *vigilin*+ cells, *globin*+ cells, *lypoxygenase*+ cells and *polycystin* cells. Next, we used graph analysis to visualise *Pristina* WGCNA modules as a network, and identified several graph connected components that reliably match the WGCNA modules and roughly recapitulate cell type-specific gene expression (Fig. 6A, Supplementary Fig. 9A, B). This allowed us to explore the relationships between gene modules by computing the number of cross-connections between pairs of modules. This highlighted connections between cilia, esophagus and *polycystin* cells suggesting the presence of cilia in such cell types (Fig. 6B), among other connections. To explore potential TFs regulating specific gene modules, we explored the centrality of TFs in each module sub-graph. We detected an agreement between TF centrality and other exploratory metrics such as TF-module connectivity (kME) (Supplementary Fig. 9C–F), revealing further putative TF regulators of each differentiated cell type including multiple homeobox, forkhead and zinc-finger TFs, among others (Fig. 6C). Overall, our analysis reveals the transcriptomic landscape of annelid adult cell type differentiation.

### *Piwi*+ cells are abundant, heterogeneous and have a pluripotent stem cell signature

Next we focused on identifying and characterising putative stem cell populations in *Pristina*. *Piwi*+ cells have been described previously in this

species[5,69] but their transcriptional profiles, cellular properties and differentiation capacities remain largely unknown. We found that the central clusters of our UMAP (1, 2 and 8) highly expressed *piwi-1* and *nanos* (Fig. 7A, left panels). These clusters constitute 21.6 % of our dataset (Fig. 1E), indicating that *piwi*+ cells are an abundant cell type in *Pristina*. The representation of *piwi*+ cells in our three independent SPLiT-seq experiments ranged from 13.0% to 33.8%. This indicates that the percentage of *piwi*+ cells is highly variable, potentially reflecting differences in the average nutritional state (and therefore growth and fission states) of worms in our three experiments. We then analysed the expression of the proliferation markers *pcna* and *mcm2*, as well as histones *h2a* and *h2b*. These genes were very highly expressed in central clusters 1 and 2 (Fig. 7A, right panels). Moreover, our PAGA analysis revealed that most differentiated cell types were connected to *piwi*+ cells by reconstructed differentiation trajectories (Figs. 1D and 7B), including epidermis, muscle and gut, suggesting these cells are a pluripotent population. While we observed expression of proliferation markers in other clusters (Fig. 7A–C), clusters 1, 2 and 8 concentrate most of their expression (ranging from 70.0% to 82.1% of all reads mapped to these features), indicating that *piwi*+ cells are the major proliferative cell type in *Pristina*. To model the developmental potency of *Pristina piwi*+ cells we calculated the potency score[18]. This graph analysis metric evaluates the normalised degree of each node of the abstracted PAGA graph as an estimation of the number of computationally predicted differentiation trajectories that connect to it. While showing the developmental potency of a cell population necessitates transplantation experiments, the potency score is a useful model to hypothesise it from single cell expression data. The highest potency score in our abstracted graph was attained by *piwi*+ cell cluster 2 (Fig. 7D), suggesting that *piwi*+ cells may be pluripotent stem cells. Clusters 16, 0, 3 and 13 also attained high potency scores, as they were connected by the PAGA analysis to several gut, neuronal, and epidermal clusters, reinforcing the scenario of them being progenitors of these differentiated types (Fig. 7D). Pluripotent cells in other organisms have been shown to be heterogeneous[20,38,70–72], consisting of mixtures of cells that co-express stem cell markers and transcripts that are characteristic of the cell types that they will differentiate into. To elucidate if *Pristina piwi*+ cells are heterogeneous we performed a subclustering of these cell clusters (Fig. 7E) and scored the markers obtained in this analysis (Fig. 7F). In this analysis, a dataset containing only *piwi*+ cells is subjected to a single cell analysis and clustering analysis to reveal subclusters of cells. *Piwi*+ subclusters contained markers of several differentiated types, including gut and epidermal cells (Fig. 7F, Supplementary Fig. 10). Furthermore, subcluster 4 showed expression of *nidogen*+ cell markers, which are connected by PAGA with muscle. These results show that *piwi*+ cells co-express stem cell markers plus markers

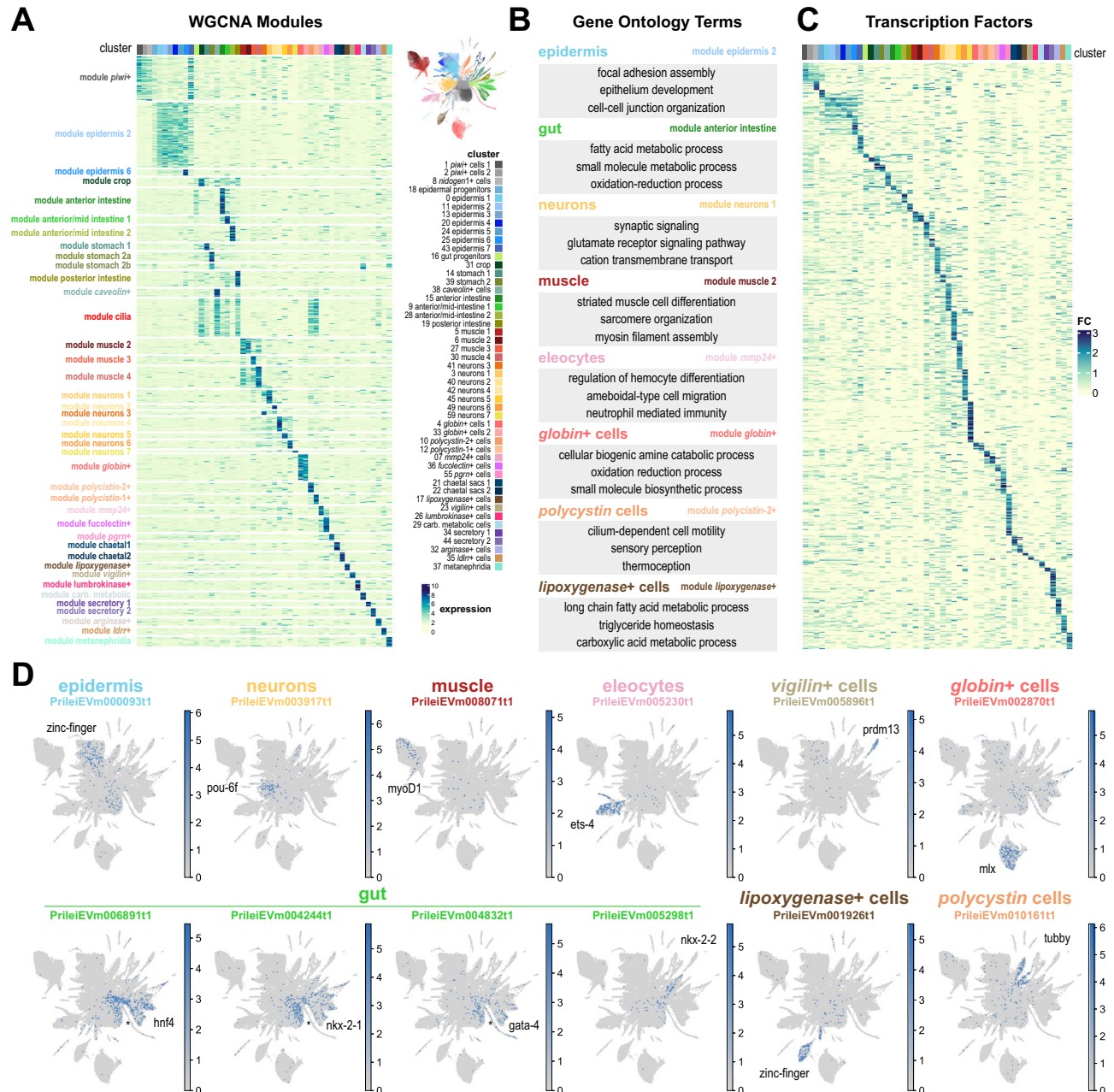

**Fig. 5 | The transcriptional landscape of annelid cell type differentiation.**
**A** Expression heatmap of 10,796 genes was classified in 40 WGCNA modules (rows), sorted by cluster expression (columns). Colour intensity indicates normalised expression (z-score). **B** Summary of Gene Ontology terms associated with example modules. **C** Expression heatmap of 650 TFs (rows) sorted by cluster expression (columns). Colour intensity indicates expression fold change. **D** Expression plots of TFs associated with individual cell types or broad types. The asterisks point to key differences between TFs.

of the several cell types that they may differentiate into. Altogether, these analyses showed that *piwi*+ cells in *Pristina* are a heterogeneous cell population with transcriptomic properties that are also observed in other pluripotent stem cells. However, individual cell potencies need to be demonstrated in future studies.

## Chromatin regulators are conserved markers of annelid *piwi*+ cells

We then sought to understand the transcriptomic profile of *Pristina piwi*+ cells. We first annotated Clusters of Orthologous Groups (COGs)[41,73] across the species transcriptome, and scored their expression in the single-cell dataset. We found that *piwi*+ cells were enriched in COGs related to chromatin, transcription, cell cycle, nuclear structure, RNA biology and DNA repair (Fig. 8A, Supplementary Data 10).

We then sought to understand their transcriptional regulation by identifying their highly expressed TFs (see Methods). Interestingly, a high proportion of TFs highly expressed in *piwi*+ cells were also highly expressed in one or more differentiated cell type groups (Fig. 8B, Supplementary Data 11, see Methods). Examples of these included TFs expressed in *piwi*+ cells and other cell types such as *vigilin*+ cells, muscle, *polycystin* cells, gut and epidermis (Fig. 8C, Supplementary Data 12). This finding is highly consistent with the specialised or lineage committed stem cell concept and suggests that these TFs are those that prime and regulate differentiation to their correspondent cell types. We then used limma (see Methods) to obtain the full transcriptional profile of *piwi*+ cells and identified a list of 735 significantly enriched transcripts (t-test with empirical Bayesian moderation of standard errors, false discovery rate by Benjamini-Hochberg,

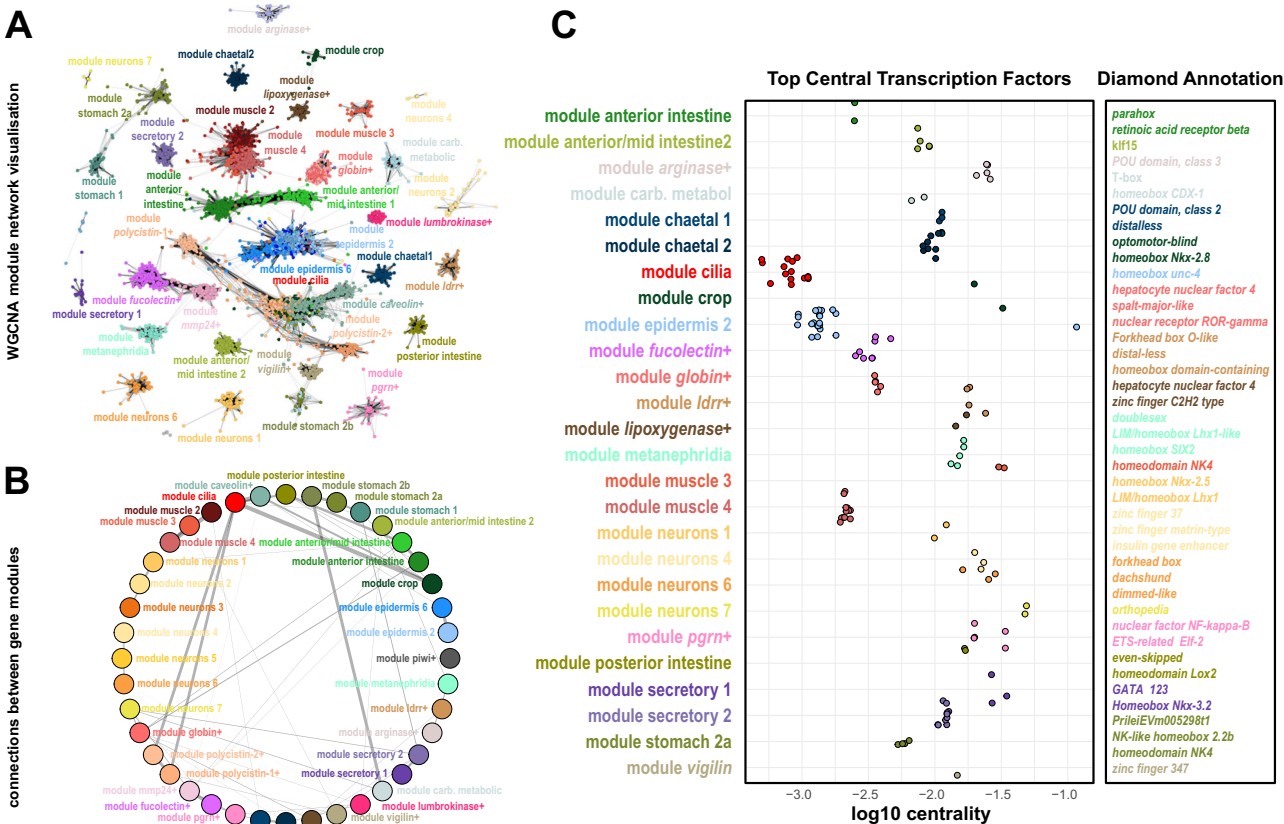

**Fig. 6 | Network analysis of *Pristina* gene modules. A** Network visualisation of WGCNA modules using the Fruchterman-Reingold layout algorithm. In this visualisation, each gene is represented as a dot, coloured according to its cluster of highest expression, and edges represent gene coexpression based on WGCNA TOM values (>0.35). 38 out of 40 modules survive this threshold (see Supplementary Note 3). **B** Module network visualisation summarising coexpression values between different modules, showing associations between different modules. Edge thickness indicates the number of co-expressed genes from different pairs of modules. **C** Stripplot showing the top central TFs identified in WGCNA modules, and their annotations.

*p*-value < 0.05, logFC > 2, Fig. 8D, Supplementary Data 13). Notably, this list included stem cell regulators such as *piwi*, *vasa*, *nanos* and *pumilio*, known to be expressed in pluripotent stem cells across the animal tree of life, as well as in germ cells[32–35]. Moreover, in the *Pristina piwi*+ cell transcriptome were cell cycle regulators, DNA repair proteins and purine synthesis enzymes, also consistent with other pluripotent stem cell transcriptomic profiles[74–77]. A very prominent feature of *Pristina piwi*+ cells was the expression of epigenetic regulators and/ or chromatin remodelers. To corroborate this feature, we used BLAST to search for homologues of the most important chromatin remodelling complex components, including the HAT, MLL, PcG, SWI/SNF, HDAC, ISWI, and FACT complexes[78,79]. We identified 156 *Pristina* transcripts encoding these (Supplementary Data 14), and found them all enriched in *piwi*+ cells (Fig. 8E). Similar to human and planarian pluripotent cells[75,76,80], this shows that high expression of epigenetic regulators is a conserved feature of animal pluripotent cells. This analysis allowed us to look for the first time at the transcriptomic features of *piwi*+ cells in annelids. Taken together, our data suggest a model where post-transcriptional and epigenetic regulators control stem cell maintenance and pluripotency, and a panoply of TFs prime these to differentiate into multiple cell types.

**In situ HCR and EdU labelling confirms that *piwi*+ cells are proliferative cells and express markers of differentiation**
We then sought to experimentally validate the proliferative properties and the heterogeneity of *piwi*+ cells. For this, we performed double in situ HCR using markers of *piwi*+ cells combined with top markers of differentiated cell types and EdU labelling of dividing cells. We chose

*histone h3* (h3, PrileiEVm022498t1) as a marker of *piwi*+ cells since i) it is one of the top markers of *piwi*+ cells (Supplementary Data 3, 4), ii) *h3*+ cells show a similar expression pattern as *piwi*+ cells, with an enrichment in the fission zone and the posterior growth zone (Fig. 9A)[5], iii) our double in situ HCR validates the coexpression of *h3* and *piwi* (Fig. 9B) and iv) the in situ HCR signal of *h3* is much stronger than *piwi*, allowing better visualisation. Double labelling of *h3*+ cells by in situ HCR and proliferating cells with EdU shows a similar distribution of the two cell populations with an enrichment in the prostomium, the fission zone and the posterior growth zone (Fig. 9A). Many of the *h3*+ cells across the body are also positive for EdU, indicating that a subset of the *h3*+ cells population is actively dividing. A portion of the EdU+ cells does not express *h3* and could be either recently differentiated cells or a lineage-restricted stem cell population.

Analysis of the single-cell dataset reveals that markers of differentiated cell types are expressed in *piwi*+ cells, like the gut marker PrileiEVm022781t1, the neuronal and *polycystin* cell marker PrileiEVm025662t1 and the epidermis marker PrileiEVm008287t1, this last one sharing orthology with intermediate filament proteins (Fig. 9C–E, Supplementary Data 1 and 3, 4). We validated colocalisation of these markers with *piwi*+ cell marker *h3* by in situ HCR (Fig. 9C–E). We observed colocalisation of *h3*, EdU and the anterior intestine marker PrileiEVm022781t1 near the anterior intestine (Fig. 9C). In the fission zone, an area enriched in actively dividing *piwi*+ cells, some *h3*+ cells express markers of differentiated cells, including neurons and *polycystin* cells (Fig. 9D) and epidermis (Fig. 9E). Interestingly, some double positive cells are also stained with EdU, highlighting either active or very recent DNA synthesis. These results validate that *piwi*+

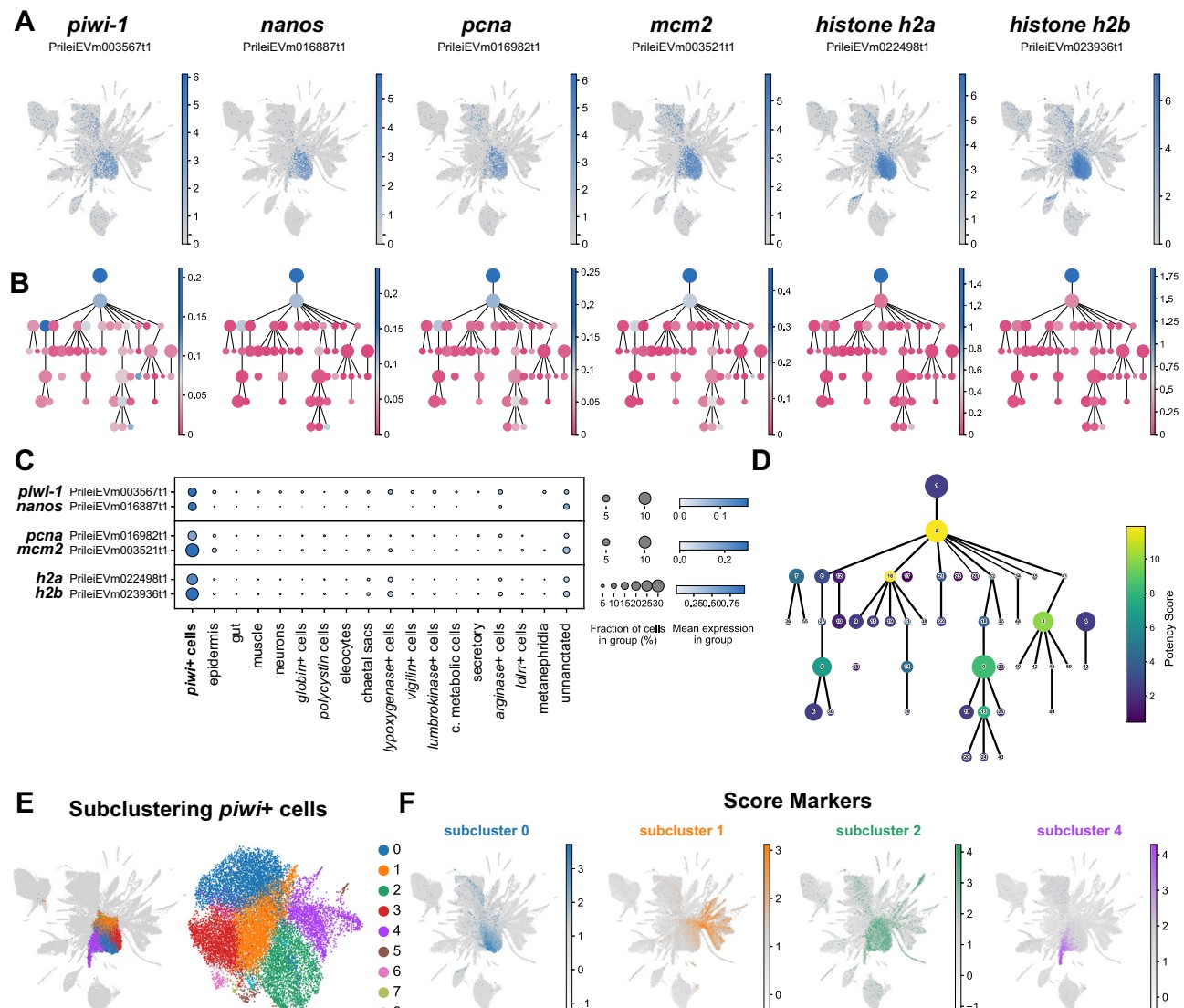

**Fig. 7 | Pluripotent stem cell signature of *Pristina piwi*+ cells. A** Expression plots of stem cell and proliferation markers: *piwi, nanos, pcna, mcm2, histone h2a,* and *histone h2b.* **B** PAGA feature plots of stem cells and proliferation markers in (**A**). The graph nodes represent the individual cell clusters and the colour intensity, from dark blue (high) to darkish pink (low), represents the expression of each marker. **C** Dot Plot showing the expression of stem cell and proliferation markers in broad cell types. The colour intensity of the dots represents the mean expression and the size of the dot represents the fraction of cells expressing the marker. Due to the

highest expression of histone genes each pair is presented in an individual panel, with its own maximum colour intensity and size. **D** PAGA plot coloured according to potency score, ranging from dark blue (low) to yellow (high). **E** UMAP visualisation of the 16,247 cells of *piwi*+ clusters 1, 2 and 8, in their original UMAP embedding (left) and their subclustering UMAP embedding (right), with clusters coloured according to their cell subcluster classification. **F** UMAP score plots of markers of *piwi*+ subclusters 0, 1, 2 and 4, showing differential expression in gut, epidermis, *lumbrokinase*+, *vigilin*+ and *nidogen*+ cell clusters.

cells are a heterogeneous cell population, with a portion of the cells coexpressing markers of at least three different lineages, and a high proliferation rate in the adult stage. Taken together, these results suggest that *piwi*+ cells in *Pristina* are actively differentiating into diverse cell types in the adult worm.

## Discussion

In this study, we report a new transcriptome and single-cell atlas of adult *Pristina leidyi*, an annelid species capable of extensive adult cell type generation and regeneration: the animal can generate all adult cell types both as part of their normal asexual growth by fission and after injury by regeneration. Our datasets provide an unprecedented perspective on adult cell type differentiation in annelids and their pluripotent cellular sources. The adult cell type atlas of *Pristina* reveals the cellular identities that make up adult annelids. We uncover ~50 distinct cell clusters and validate many of them using a newly

developed multiplexed in situ HCR approach. Our data reveal well-known cell types such as epidermis and muscle, a complex organisation of the annelid gut, as well as multiple annelid-specific cell types and novel cell types. We studied their distribution patterns along the body as well their transcriptional and regulatory profiles, including gene expression modules and transcription factors. These new cell types offer key information to the field of cell type evolution, a field that has been reinvigorated by single cell transcriptomics. For instance, we found a *vigilin*+ cell type that expresses mucins and is localised in the head region, indicating that these are *Pristina* pharyngeal glands, previously described in other oligochaeta species. Interestingly, *vigilin* has been implicated in polyploidisation events[57] and we show that *vigilin*+ nuclei have larger sizes, consistent with a plausible polyploidisation. Nevertheless, further analyses would be necessary to confirm our hypothesis and to elucidate the function of this cluster.

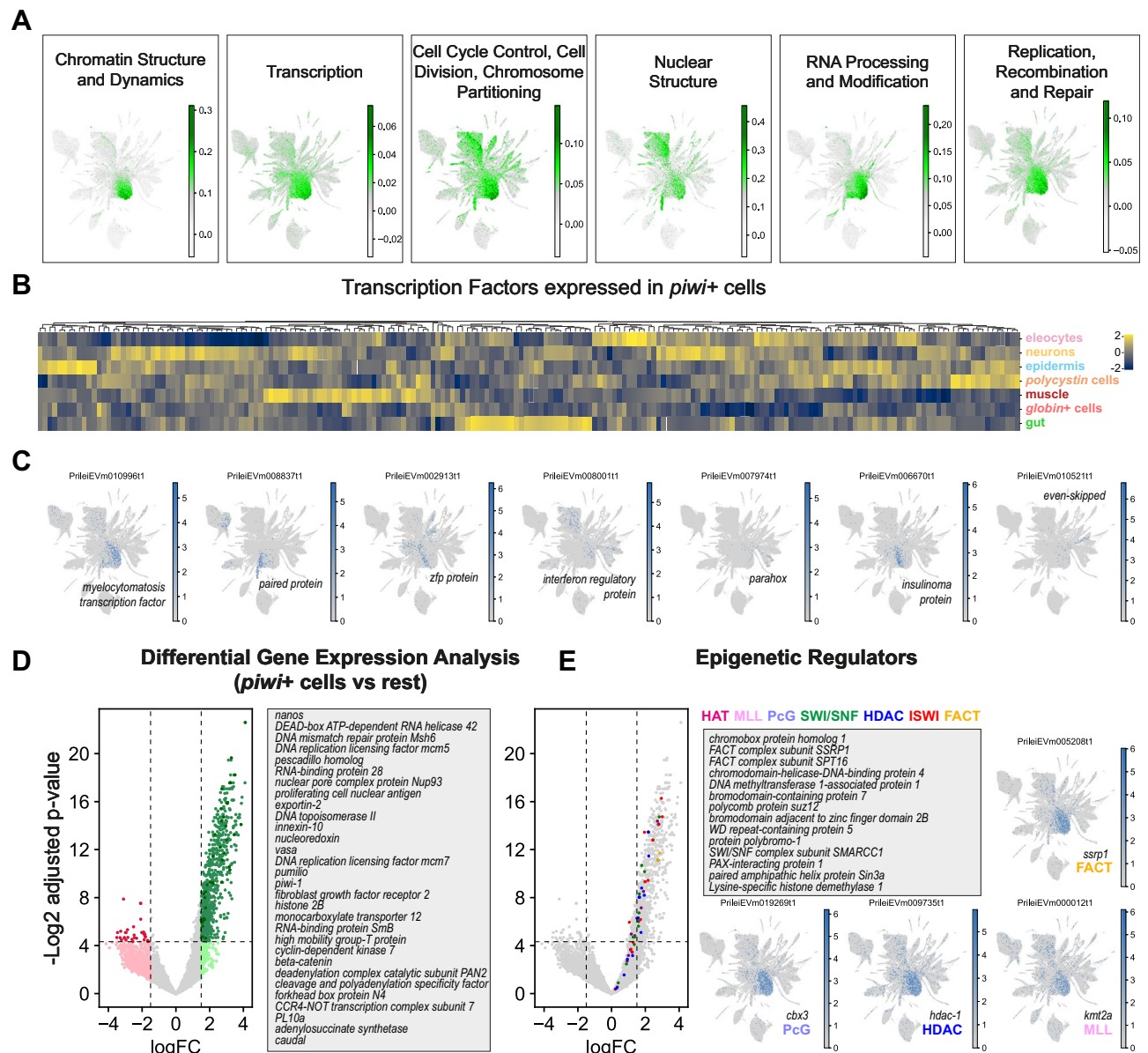

**Fig. 8 | Transcriptomic profile of annelid *piwi*+ cells. A** UMAP visualisation of scored gene expression of COGs in *piwi*+ cells. **B** Expression heatmap of 200 top TFs expressed in *piwi*+ cells and their expression in the main broad cell types, showing several clusters of TFs expressed in both *piwi*+ cells and one or more broad types. **C** Expression plots of example TFs coexpressed in *piwi*+ cells and other broad types. **D** Limma differential gene expression analysis of *piwi*+ cells (clusters 1, 2 and 8) against all other cell types (bayesian t-statistics from the eBayes limma function, two-sided; adjusted *p*-values with Benjamini-

Hochberg correction). Green colour indicates upregulated genes, red colour indicates downregulated genes. Light colour shade indicates above threshold of logFC, darker colour shade indicates above threshold of logFC and significant (adjusted *p*-value < 0.05). Examples listed are coloured in darker green. **E** Detail of annotated epigenetic regulators and their expression enriched in *piwi*+ cells, and example UMAP visualisations of representative epigenetic factors.

Cell types such as the *globin*+ cells and the eleocytes could be representatives of blood types related to haemocytes in other species and vertebrate blood cells. On the other hand, cell populations such as the *ldlrr*+ cells and the carbohydrate metabolic cells have no known homologue cell types in other groups. Future studies will focus on transcriptomic comparisons of these cell types to elucidate their evolution.

The differentiation of the majority of these cell types can be reconstructed from the *piwi*+ cell population in *Pristina*, which shows hallmarks of pluripotency. First, it expresses conserved RNA-binding proteins such as *vasa*, *nanos*, *pumilio* and *piwi*. These transcripts have been found in pluripotent stem cells in sponges, cnidarians, acoels, planarians, colonial ascidians and other organisms, as well as

the germ line of most animals[32–35]. Second, differentiation trajectories from *piwi*+ cells to a broad collection of cell types can be computationally reconstructed using lineage reconstruction algorithms[16,48]. These exploit the presence of cells captured along their differentiation process, with transcriptomes intermediate between those of stem cells and differentiated cells. The concept of germ layers is key to the definition of pluripotency, but it is difficult to apply to asexually reproducing animals, where all cell types are differentiated from adult populations rather than embryonic germ layers. We therefore apply the pluripotency definition based on the reconstructions to broadly different cell types, including epidermis, muscle and gut, known to originate from distinct embryonic germ layers in annelids[81–86]. Third, the *piwi*+ cell cluster is

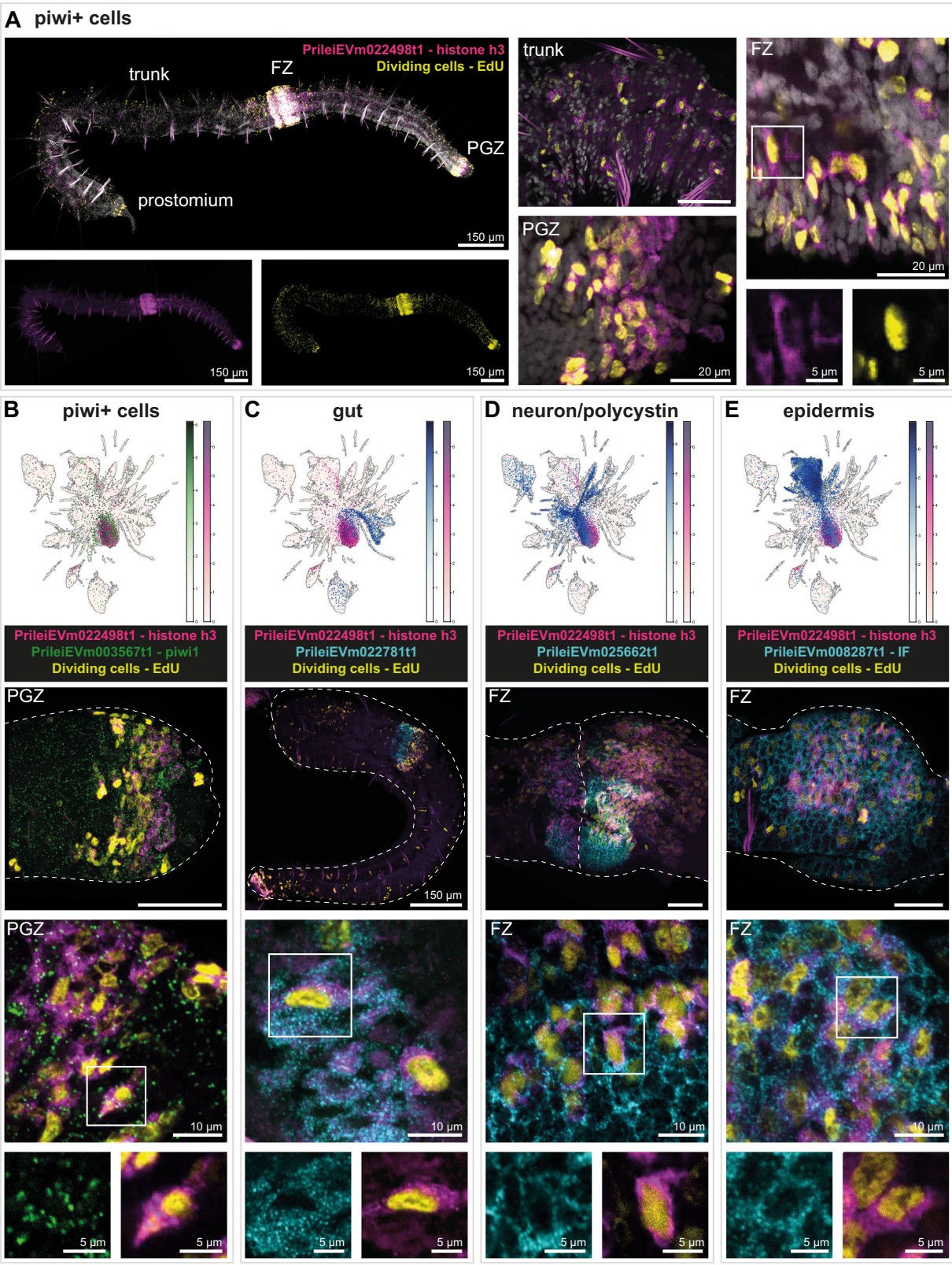

heterogeneous and includes subpopulations that express stem cell markers and markers of differentiation to broad cell type groups or individual types. This is consistent with the idea of lineage committed stem cells that have already started their differentiation process[20,38,70–72]. Our analysis reveals rich regulatory information, including dozens of transcription factors that are expressed in *piwi*+ cells and in a given set of differentiated types. Fourth, our analysis

uncovers a high expression of epigenetic regulators and chromatin remodelers in *piwi*+ cells. Many epigenetic regulation complexes are expressed in *piwi*+ cells at levels higher than those observed in differentiated cells. This is a signature of pluripotency in human[87,88] and planarian stem cells[76,77], but is still understudied in other models. Importantly, *piwi*+ cells concentrate most of the expression of cell cycle related transcripts but we cannot rule out that other cell types

**Fig. 9 | In situ HCR expression of proliferation and differentiated cell markers in *piwi*+ cells. A** In situ HCR expression of *piwi*+ cells marker PrileiEVm022498t1 (*histone h3*, magenta) and EdU+ cells (yellow) showing signal throughout the whole animal body. The right microscopy panels are close-ups from different animals, showing overlapping expression in the trunk, fission zone (FZ), and posterior growth zone (PGZ). The bottom right microscopy panel is a close-up from the upper right microscopy panel, evidencing the overlapping expression in a cell in the FZ. All cells were stained with DAPI (grey). **B** In situ HCR and expression plot of *piwi*+ markers PrileiEVm022498t1 (*histone h3*, magenta) and PrileiEVm003567t1 (*piwi1*, green), and EdU+ cells (yellow), showing extensive signal in the PGZ. The middle and bottom microscopy panels are close-ups from the upper microscopy panel evidencing overlapping expression in the PGZ and at the cellular level. Dashed line indicates the outline of the worm. **C** In situ HCR and expression plot of *piwi*+ marker PrileiEVm022498t1 (*histone h3*, magenta) and gut marker PrileiEVm022781t1 (cyan), and EdU+ cells (yellow), showing expression in the developing gut of the new worm that has just split apart. The middle and bottom microscopy panels are close-ups from the upper microscopy panel evidencing overlapping expression in the gut and at the cellular level. **D** In situ HCR and expression plot of *piwi*+ marker PrileiEVm022498t1 (*histone h3*, magenta), and neural and polycystin marker PrileiEVm025662t1 (cyan), and EdU+ cells (yellow), showing intensive expression in the developing brain in FZ. The middle and bottom microscopy panels are close-ups from the upper microscopy panel evidencing overlapping expression in the developing brain and at the cellular level. **E** In situ HCR and expression plot of *piwi*+ marker PrileiEVm022498t1 (*histone h3*, magenta) and epidermal marker PrileiEVm008287t1 (*intermediate filament*, cyan), and EdU+ cells (yellow), showing intensive expression in the FZ. The middle and bottom microscopy panels are close-ups from the upper microscopy panel evidencing overlapping expression in the epidermis and at the cellular level. In all panels, anterior is left, dorsal is up. Scale bars are 50 μm unless noted in the figure. All expression patterns displayed in the figure were observed in, at least, three different individuals.

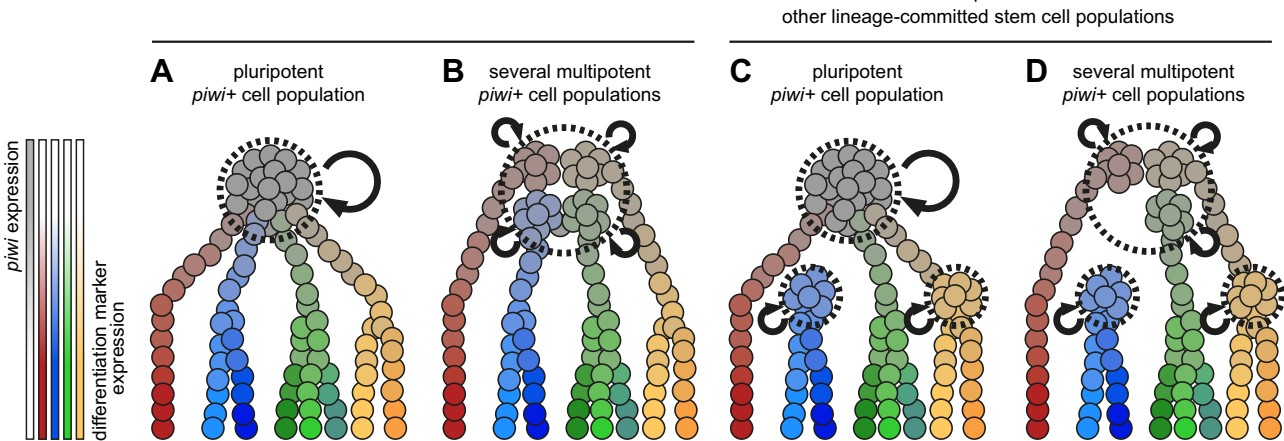

**Fig. 10 | Alternative hypotheses of stem cell function in *Pristina* adult cell type generation. A–D** Diagram of 4 alternative models of stem cell function in *Pristina*. *Piwi*+ cells are depicted in grey, and differentiation markers are depicted in red, blue, green and yellow. Stem cell populations are shown within dashed lines and self-renewal is represented by curved arrows.

are able to undergo cell division. For instance, some epidermal clusters also express cell proliferation markers and histones. The expression of epigenetic regulators is however very restricted to *piwi*+ cells.

Our data reveal a prominent *piwi*+ cell population in *Pristina* and allows us to hypothesise its pluripotent nature, but this aspect remains to be experimentally validated by direct methods. There are several possibilities: individual *Pristina piwi*+ cells could be pluripotent, and could be the only stem cells in the adult (Fig. 10A). This scenario is very difficult to distinguish from an alternative scenario, where several lineage-committed *piwi*+ stem cell populations coexist and are indistinguishable by our single cell transcriptomic data (Fig. 10B). Another possibility is that other lineage-committed stem cell populations exist, but are *piwi* negative (Fig. 10C). These could be lineage related to *piwi*+ cells or be an independent lineage. The expression of proliferation markers in the epidermis cluster, together with the observed EdU incorporation in the epidermis (Fig. 9A), suggests that epidermal stem cells might exist in *Pristina*. However, further work is needed to determine if these epidermal cells are *piwi*+, if they are a stem cell population capable of self-renewal and if they constitute a niche isolated from the main *piwi*+ stem cell pool. Finally, a combination of several scenarios is also possible (Fig. 10D). Altogether, our study reveals a *piwi*+ cell population with the hallmarks of pluripotency and suggests that it underlies adult cell type generation in posterior growth and fission in annelids.

# Methods

## *Pristina leidyi* culture and maintenance

*Pristina leidyi* culture was originally obtained from Carolina Biological Supply[89]. Specimens were cultured both in plastic boxes and fish tanks with 1 L and 50 L of 1% filtered artificial seawater, respectively. Water was changed every week and animals were fed with 0.03 g/L of dried spirulina powder every 2 weeks. Under these conditions, worms reproduce continuously by paratomic fission[89]. No ethical approval was required to work with annelids.

## Iso-seq

Approximately 100 *Pristina leidyi* of mixed conditions, including fissioning animals, were manually picked out of culture using a glass Pasteur pipette. These were placed into a single 1.5 mL Eppendorf tube, and spun on a low speed benchtop centrifuge to pellet. The supernatant was removed. Total RNA was extracted from the pelleted worms using the Trizol method and the standard manufacturer protocol. The quality of this was assessed using a Nanodrop, giving a concentration of 1083.7 ng/uL, an A260/A280 ratio of 2.01 and an A260/A230 ratio of 2.03. Quality was further assessed using a Bioanalyzer (Agilent), although a RIN value was not calculated due to the difference in profile commonly observed in annelid RNA samples. Total RNA was provided to the Earlham Institute Genomics Pipelines Group, Norwich, UK, and (after QC to confirm quality) was used as the basis of PacBio Iso-Seq Express Template Preparation (v2) library construction. This sample,

along with 3 others, was loaded onto a PacBio Sequel II SMRT cell, and sequenced (8 M, v2, 30 h Movie).

Iso-Seq3 analysis was performed by the provider. A total of 3,932,103 CCS reads were captured across the samples on the cell, with 1,546,939 assigned to *Pristina leidyi*. These were classified and clustered, resulting in 54,350 high-quality isoforms.

## Sequence concatenation and redundancy removal

The sequences gained from Iso-Seq sequencing analysis were combined with sequences derived from previous analysis of the *Pristina leidyi* transcriptome[90]. First, the isotigs from the Nyberg et al. dataset were concatenated with the isoform sequences derived from Iso-Seq analysis. Redundancy was removed from these reads using the EvidentialGene[91] tr2aacds4.pl approach (March 2020 v4 version) with settings -cdnaseq -NCPU 8 -MAXMEM 16000 -logfile, keeping only a single sequence representative per locus with the best evidence score. Transdecoder v5.5 was then used to predict the protein coding regions of transcripts (LongOrfs -m 25, Predict --single_best_only).

## Diamond Blast annotation

We implemented diamond v2.0.8.146[42,43] to provide an initial putative identity to orthologs present in our reference transcriptome. This software performed a blastx search against the whole downloaded database with default settings and organised the results into a table with the settings --salltitles -b8 -c1 -p8 --outfmt 6 qseqid sseqid pident evalue stitle.

## eggNOG annotation

The assembled transcriptome of *Pristina leidyi* was transformed to protein sequence using TransDecoder (https://github.com/TransDecoder/TransDecoder/wiki); first, we ran 'TransDecoder.LongOrfs' with standard parameters; second, we ran hmmscan vs Pfam database and BLAST vs Swissprot database, with parameters:'-max_target_seqs 1 -evalue 1e-5' and default parameters respectively, to gather supporting evidence for coding transcripts; third, we ran 'TransDecoder.Predict' with parameters '--retain_pfam_hits pfam.domtblout --retain_blastp_hits blastp.outfmt6 --single_best_only'. The resulting translated transcriptome (hereafter referred to as proteome) was queried using EggNOG mapper[41] with the parameters: '-m diamond --sensmode sensitive --target_orthologs all --go_evidence non-electronic' against the EggNOG metazoa database. From the EggNOG output, GO term, functional category COG, and gene name association files, were generated using custom bash code. Full code is available at the project repository.

## ACME dissociation

Our data comprises three different replicated experiments (batches) with independently sourced worms from different ACME dissociation samples. Depending on the experiment, animals were not fed for: 12 days (library 12), 4 days (library 21) or 7 days (library 30). ACME was performed as previously described[44] with some modifications. For each sample, we added ~120 *Pristina leidyi* worms at mixed stages (including fissioning animals) to a 15 mL Falcon tube (~100 uL of biomass volume). Sex was not determined, as *Pristina* does not sexualise in lab conditions. We removed most culture water and added 300 uL of NAC solution per tube. NAC solution was freshly prepared by diluting N-acetyl cysteine powder in 1x PBS buffer to a 7.5% w/v. The 1x PBS buffer was made from a nuclease-free 10x PBS stock solution. We flicked samples in NAC for 30", and added 10 mL of ACME solution per tube immediately after. The ACME solution was prepared fresh using 6.5 mL of nuclease-free H2O, 1.5 mL of methanol, 1 mL of acetic acid and 1 mL of glycerol per sample. Samples were incubated in ACME for 35 min, at room temperature, in a rocking table (40–45 rpm). To help dissociation, tubes were manually shaken every 10 min. After incubation, samples were pipetted up and down to complete dissociation.

From this point, samples were kept on ice to prevent RNA degradation. With cells still on ACME, we filtered through 50 μm strainers (CellTrics) into new 15 mL Falcon tubes. Samples were centrifuged at 1000 g for 6 min (4 °C) to remove ACME, and pellets were resuspended in 8 mL of 1x PBS 1% BSA fresh buffer. We centrifuged again at 1000 g for 6 min (4 °C) and discarded the supernatant. Pellets were resuspended in 900 uL of 1x PBS 1% BSA (Thermo Fisher, cat. BP9700100) fresh buffer and transferred to 1.5 mL Eppendorf tubes. To cryopreserve cells, we added 100 μL of DMSO per sample and stored at −80 °C.

## SPLiT-seq

All oligonucleotide sequences used in this protocol are the same as those used in García-Castro et al.[44]. SPLiT-seq was performed as previously described[44] with the following modifications:

Cell count: Cryopreserved ACME-dissociated cells were thawed and centrifuged twice at 1000 g for 6 min (4 °C) to remove the DMSO. Pellets were resuspended in 250 uL of 1x PBS 1% BSA fresh buffer. For each sample, we prepared a separate 1:3 dilution with 50 uL of cells and 100 uL of buffer. Dilutions were stained for 15 min, at RT, with 0.2 uL of DRAQ5 (5 mM stock solution, Bioscience, cat. 65-0880-96) and 0.6 uL of Concanavalin-A conjugated with AlexaFluor 488 (1 mg/mL stock solution, Invitrogen, cat. C11252). The remaining undiluted samples were kept at 4 °C. Cell count was performed on the stained dilutions by flow cytometry. From this, we calculated the concentration on the main samples and diluted them to a final working concentration of 625–1250 events/uL.

**Round 1 of barcoding: reverse transcription.** The Round 1 plate was loaded with 8 uL/well of Round 1 barcodes, 8 uL/well of cells at a concentration of 625-1,250 events/uL (5,000-10,000 events per well) and 8 uL/well of the following RT mix: 4 μL of 5x Maxima RT Buffer (Thermo Scientific, cat. EP0753), 0.375 μL of Superase-In RNAse inhibitor (20 U/μL, Invitrogen, cat. AM2696), 1 μL of 10 mM/each dNTPs (NEB, cat. N0447S), 0.625 μL of nuclease-free H2O and 2 μL of Maxima H Minus RT (200 U/μL, Thermo Scientific, cat. EP0753). In library 30, we also added 10% w/v of PEG 8000 to the RT mix. The reverse transcription reaction ran in a thermocycler for 35 min at 50 °C. After incubation, reactions were pooled in a 15 mL Falcon tube. We added 10% Triton X-100 to the cells, to a final concentration of 0.1%, and centrifuged at 1200 g for 6 min. Cells were resuspended in 2 mL of NEBuffer 3.1 (NEB, cat. B6003S) with 20 uL of Superase-In RNase Inhibitor.

**Round 2 of barcoding: ligation 1.** The ligation mix was prepared with 500 μL of 10x T4 Ligase Buffer, 100 μL of T4 DNA ligase (400 U/μL, NEB, cat. M0202L), 100 μL of 1x PBS 1% BSA buffer, and 1340 μL of nuclease-free water. For library 30, we additionally added 10% w/v of PEG 8000 to the ligation mix.

**Round 3 of barcoding: ligation 2.** Pooled cells from Round 2 were mixed with 150 μL of T4 DNA ligase. The Round 3 plate was loaded with 55 μL/well of this mix.

**Washing.** After last blocking, we pooled cells in a 15 mL Falcon tube and added 10% Triton-X 100 to a final concentration of 0.1%. Cells were centrifuged at 1200 g for 6 min (4 °C). The supernatant was discarded and the pellet was resuspended in 4.04 mL of washing buffer (4 mL of 1x PBS and 40 μL of 10% Triton X-100). Cells were centrifuged again, resuspended in 800 uL of 1x PBS 1% BSA buffer, and split in two 1.5 mL Epp tubes (400 uL/each). These samples were stored at −80 °C in 10% DMSO.

**FACS.** FACS was performed in the middle of the SPLiT-seq protocol. We thawed previously barcoded samples, added 2 μL of 10% Triton X-100 per tube, and centrifuged at 1200 g for 6 min (4 °C) to eliminate

the DMSO. Supernatants were carefully discarded, and pellets were resuspended in 500 μL of 1x PBS 1% BSA buffer. We added another 2 uL of 10% Triton X-100 per tube and repeated centrifugation in the same conditions. Final pellets were resuspended in 400 μL of 1x PBS 1% BSA buffer and stained with 0.5 μL of DRAQ5 and 1 μL of Concanavalin-A conjugated with AlexaFluor 488. Stained cells were incubated for 45 min, on ice, in a dark box. Cells were sorted using a BD FACS Aria III (BD Biosciences) set in 4-ways Purify Mode and 45 Psi of pressure, with an 85-um nozzle. DRAQ5 and Concanavalin-A positive singlets were sorted in sub-libraries of 9000-25,000 cells, collected directly into 50 uL of 2x Lysis Buffer. FACS time was about 1.5 hours per batch.

**Cell lysis.** The sorted sub-libraries were adjusted to a volume of 100 uL, when necessary, using 1x PBS 1% BSA buffer. We added 10 μL of Proteinase K (20 mg/mL, Thermo Fisher, cat. EO0491) to each sub-library and incubated for 2 h at 55 °C. After incubation, lysates were frozen at −80 °C.

**Template switch.** The Template Switch mix was prepared using 44 μL of 5x Maxima RT Buffer, 44 μL of 20% Ficoll PM 400 (Sigma Aldrich, cat. GE17-0300-10), 22 μL of 10 mM/each dNTPs, 5.5 μL of Superase-In RNASe inhibitor, 5.5 μL of TSO primer (100 μM), 11 μL of Maxima H Minus RT (200 U/μL), 0.022 g (10% w/v) of PEG 8000 (only for libraries 21 and 30), and up to 220 μL of nuclease-free water per sample.

**PCR amplification.** Samples were amplified for 5 cycles of PCR and 10-11 cycles of qPCR.

**Size selection.** We purified qPCR reactions by two consecutive rounds of SPRI size selection at ratios of 0.8x and 0.7x. After the first 0.8x size selection, the eluted volume (20 uL) was adjusted to 100 uL using nuclease-free water. Final fragment distributions and concentrations were assessed by running a High Sensitivity DNA bioanalyzer (Agilent 2100, cat. 5067-4626) and a Qubit dsDNA High Sensitivity Assay (Thermo Fisher, cat. Q32851), respectively, according to the manufacturer's protocols.

**Tagmentation.** Tagmentation was performed using the Nextera XT DNA Library Preparation Kit (Illumina, cat. FC-131-1024). We prepared the tagmentation reactions by mixing 5 μL of cDNA (1 ng in total), 10 μL of Tagment DNA Buffer (TD) and 5 μL of Amplicon Tagment Mix (ATM). Reactions were incubated in a preheated thermocycler for 5 min at 55 °C. Samples were placed on ice immediately after incubation. To stop tagmentation, we added 5 μL of Neutralize Tagment Buffer (NT), mixed well, and incubated at room temperature for 5 min.

**Round 4 of Barcoding: PCR.** We prepared a separate reaction mix for each sub-library, containing 22 μL of tagmented cDNA, 15 μL of Nextera PCR Master Mix (Nextera XT DNA Library Preparation Kit), 1 μL of P5_oligo (10 μM) and 1 μL of a Round 4 barcode (10 μM). We used different barcodes for each sub-library. The PCR reaction ran as follows: 72 °C (3 min); 95 °C (30 s); 12 cycles of 95 °C (10 s), 55 °C (30 s) and 72 °C (30 s); and 72 °C (5 min). PCR samples were purified by two subsequent rounds of SPRI size selection (0.7x and 0.6x). Fragment distribution was assessed running a High Sensitivity DNA bioanalyzer and final concentrations were quantified using a Qubit dsDNA High Sensitivity Assay.

## SPLiT-seq read processing

SPLiTseq reads were provided by Novogene (China). A total of 124,349,078 (12_1), 135,900,060 (12_2), 410,765,606 (21_1), 833,784,688 (21_2), 807,486,658 (21_3), 643,285,668 (30_2), 627,640,824 (30_3), 711,569,074 (30_4), 725,038,254 (30_5) reads were sequenced. These were assayed for QC purposes using FastQC (https://www.bioinformatics.babraham.ac.uk/projects/fastqc/, v0.11.9, 2019) and

residual adaptor sequence, low-quality, and short reads were observed. CutAdapt v2.8[92] was used to trim read 1 (transcripts) and read 2 (UMI and barcodes) sequences. The following settings: cutadapt -j 4 -m 60 -q 10 -b AGATCGGAAGAG were run for read 1. To trim read 2, settings: cutadapt -j 4 -m 94 --trim-n -q 10 -b CTGTCTCTTATA were used. To confirm barcodes were correctly in position, and not affected by indels, read 2 sequences were checked for "phase" using grep, with known flanking sequence as a search. Reads were retained when UMI and UBC barcodes were in the correct location. Finally, pairfq make-pairs v 0.17 (https://github.com/sestaton/Pairfq) was used to retain correctly paired, complete reads. These were fed into SPLiTseq tool-box (https://github.com/RebekkaWegmann/splitseq_toolbox.v 1.0) for further analysis.

The Iso-seq transcriptome of *Pristina leidyi* assembled as described above was created to have a reference database for read mapping. We then used Dropseq_tools-2.3.0 (https://github.com/broadinstitute/Drop-seq/releases/tag/v2.3.0) to process the generated GTF file and create a sequence dictionary, a refFlat, a reduced GTF and the corresponding interval files. We generated a reference index using STAR-2.7.3a[93] with the parameters --sjdbOverhang 99 --genomeSAindexNbases 13 --genomeChrBinNbits 14. Each of the sub-libraries was processed separately and properly combined later in the analysis. The SPLiTseq toolbox (https://github.com/RebekkaWegmann/splitseq_toolbox which envelops algorithms from Drop-seq_tools-2.3.0, was used to retrieve, correct and label the barcodes with a hamming distance ≤1. Mapping to the reference transcriptome used STAR-2.7.3a https://github.com/alexdobin/STAR/releases/tag/2.7.3a) with --quantMode GeneCounts and all other default settings with the exception of --outFilterMultimapNmax 10 to retain and analyse reads which mapped up to ten different loci in the reference. We implemented Picard v2.21.1-SNAPSHOT (https://github.com/broadinstitute/picard) to re-order, merge, align and tag reads for each sub-library with the SortSam and MergeBamAlignment features. We implemented sequentially the features Drop-seq_tools-2.3.0 TagReadWithInterval and TagReadWithGeneFunction to create expression matrices of each library with the feature of Drop-seq_tools-2.3.0 DigitalExpression with the settings: READ_MQ = 0, EDIT_DISTANCE = 1, MIN_NUM_GENES_PER_CELL = 50, and LOCUS_FUNCTION_LIST = INTRONIC. These matrices together with the gene models and raw reads are uploaded to GEO under the accession code GSE230505.

## Doublet identification and analysis

We used Scrublet[46] to identify potential doublets. We used the implementation in the Scanpy package[94], with the 3 different experiments as "batch keys" and an empirically optimised threshold of 0.14. With these conditions, Scrublet classified as doublets 2870 of the 80,387 cell barcodes. To independently identify doublets, we implemented a deep learning model with Solo 0.1[47]. We trained the model with default settings except for a maximum number of 400 epochs. After subsetting the calculated doublet scores per cell, we filtered by the top putative doublets (>1.5). Full code implemented is available at the project repository.

We then preprocessed this dataset containing doublets to analyse their effects in cell clusters. This dataset contains 80,387 cells, of which 2870 and 2554 cells were considered doublets by Scrublet and Solo respectively with a 458 overlap. The processing eliminated genes with high counts using sc.pp.filter_genes with max_counts = 1000000. Then we calculated metrics using sc.pp.calculate_qc_metrics, sliced the matrix genes_by_counts <700 and total_counts <900, and normalised the matrix using sc.pp.normalize_total with a target_sum=1e4. We selected high variable genes using sc.pp.highly_variable_genes with n_top_genes = 18000, and sliced the matrix to contain only those genes, storing the raw in an adata.raw object. We then scaled the matrix with sc.pp.scale, performed pca with sc.tl.pca, constructed a kNN graph with sc.pp.neighbours, with 45 neighbours and 105

principal components, and calculated a UMAP visualisation with sc.tl.umap. We then plotted doublet cells identified by scrublet, solo and both in this visualisation. To determine if these doublets were major contributors to cell clusters, we run a clustering algorithm using sc.tl.leiden with resolution parameters 1, 2, 3 and 4. These gave respectively 47, 70, 83 and 89. We then calculated the proportions of doublets in each cluster using pandas and plotted them using matplotlib.

## Parameter space optimisation

We optimised the parameter space iteratively running a custom function that processes the dataset accepting different arguments (minimum genes counts, maximum number of genes, maximum number of counts, number of top highly variable genes, number of neighbours, number of principal components, and leiden clustering resolution) and saves a figure report. The figure report includes a number of informative genes identified from preliminary analyses of the dataset because of their specific but also relatively complex expression pattern (PrileiEVm023936t1, PrileiEVm008309t1, PrileiEVm011741t1, PrileiEVm021316t1, PrileiEVm022250t1, PrileiEVm000325t1, PrileiEVm013699t1, PrileiEVm020595t1), as well as the UMAP visualisation and the number of clusters obtained. This function was run on the 75,421 cell dataset with the doublets excluded. We sequentially run iterations of this function trying the following values: minimum genes counts (30, 40, 50, 60, 70, 80, 90, 100), maximum number of genes (300, 400, 500, 600, 700, 800, 900, 1000), maximum number of counts (500, 600, 700, 800, 900, 1000, 1100, 1200), number of top highly variable genes (4000, 6000, 8000, 10000, 12000, 14000, 18000, 22000), number of neighbours (15, 25, 35, 45, 55, 65, 75, 85), number of principal components (15, 25, 45, 65, 85, 105, 125, 145), with the other parameters in each iteration remaining fixed in standard values (50, 700, 900, 18000, 45, 105, 1 respectively). We examined the result of each run to visually inspect the complexity of the cluster visualisation and the number of clusters obtained.

## Single cell transcriptomic analysis

We processed the final dataset with conditions optimised from our parameter space exploration. We started this processing with the matrix of 75,421 cells after doublet exclusion. The processing eliminated genes with high counts using sc.pp.filter_genes with max_counts = 1000000. We calculated metrics using sc.pp.calculate_qc_metrics, sliced the matrix genes_by_counts <700 and total_counts <900. This step eliminated 203 cells, giving us our final dataset of 75,218 cells. We normalised the matrix using sc.pp.normalize_total with a target_sum=1e4. We selected high variable genes using sc.pp.highly_variable_genes with n_top_genes = 18000, and sliced the matrix to contain only those genes, storing the raw in an adata.raw object. We then scaled the matrix with sc.pp.scale, performed pca with sc.tl.pca, constructed a kNN graph with sc.pp.neighbours, with 45 neighbours and 105 principal components, and calculated a UMAP visualisation with sc.tl.umap (min_dist=0.5, spread = 1, alpha = 1, gamma = 1.0). We run the Leiden clustering algorithm using sc.tl.leiden with resolutions 0.5, 1, 1.5 and 2, which gave 34, 50, 60 and 70 clusters respectively. We calculated marker genes for each cluster using sc.tl.rank_genes_groups, using the clusters of obtained with all 4 resolution parameters, and using both the Wilcoxon (method = 'wilcoxon') and the Logistic Regression (method = 'logreg') We selected resolution 1.5 for further downstream analyses.

## PAGA

For the PAGA analysis we removed unannotated clusters. Preliminary analyses indicated that these small clusters interfere with the PAGA analysis. The expression of *piwi* in them is relatively high, suggesting that they could be subpopulations of *piwi*+ cells,

but they also had specific markers, suggesting that they contain differentiated types. Our interpretation of these clusters is that they are rare cell types that, at this resolution, are clustered together with their progenitor including *piwi*+ cells. The presence of these confounds the PAGA analysis. Alternatively, they could represent leftover doublets. Altogether they are a small number of cells. To identify these clusters we calculated the mean of each transcript from the adata.X object and ranked the expression of stem cell genes by obtaining the average mean expression of PrileiEVm016887t1, PrileiEVm004300t1, PrileiEVm003567t1, PrileiEVm016982t1, and PrileiEVm003521t1. This generated a rank of clusters that contained *piwi*+ cells including clusters 1, 2 and 8 (with 7103, 6557 and 2587 cells) but also contained smaller clusters with ~2 orders of magnitude fewer cells, including clusters 51, 57, 58, 48, 43, 53, 52, 50, 47 (with 85, 50, 41, 153, 191, 74, 77, 117 and 154 cells). We decided to leave unannotated clusters with ranked expression > 0.0500 and fewer than 175 cells, which gave us the final list of clusters 46, 47, 48, 50, 51, 52, 53, 54, 56, 57, and 58.

We then performed a PAGA analysis with and without these clusters. We selected a random cell from cluster 1 as roon using adata.uns['iroot'] = np.flatnonzero(adata.obs[clusteringlayer] == '1')[0] We then used the Scanpy implementation of Difussion Pseudotime, using sc.tl.dpt(adata, n_branchings=1). We then run sc.tl.paga on the selected clusters of resolution 1.5. Our PAGA plot is generated with sc.pl.paga(adata, threshold=0.25, solid_edges = ' connectivities_tree', root=1, layout = 'rt', node_size_scale=2, node_size_power = 0.9, max_edge_width = 3, fontsize = 20). The Potency Score was plotted using sc.pl.paga with similar parameters and passing colour = 'degree_solid', cmap = 'viridis' arguments to the function.

## CPM calculation

Raw UMI counts were extracted with a custom Python script (see project repository) that slices the raw unprocessed matrix to contain only the cells that are present in the processed matrix. The cluster information is transferred from the processed matrix to the unprocessed matrix using a pandas script. Then the sum of all counts for each gene in each cluster is obtained using numpy on the matrix. The resulting raw summed counts dataset was normalised by pseudobulk "library size" using the 'DESeqDataSetFromMatrix()' function with parameter 'design = ~ condition' and the 'counts()' function with parameter 'normalised = TRUE' from the package DESeq2[95].

## Co-occurrence analysis

Cell type co-occurrence analysis was performed using the function 'treeFromEnsembleClustering()' from the code provided by Levy and collaborators[49] using parameters: 'h = c(0.75,0.95), clustering_algorithm = "hclust", clustering_method = "average", cor_method = "pearson", p = 0.1, n = 1000, bootstrap=FALSE'. Briefly, we performed 1000 iterations of cross-cell type Pearson correlation using 90% downsampling of highly variable genes (FC > 1.5) followed by hierarchical clustering of cell types. Co-occurring pairs of cell types across iterations are quantified to generate a co-occurrence matrix that is hierarchically clustered to generate the cell type tree.

## Transcription factor annotation

The resulting TransDecoder-translated proteome of *Pristina* was queried for evidence of Transcription Factor (TF) homology using (i) InterProScan[96] against the Pfam[97], PANTHER[98], and (ii) SUPERFAMILY[99,100] domain databases with standard parameters, (iii) using BLAST reciprocal best hits[101] against swissprot transcription factors[102], and (iv) using OrthoFinder[103] with standard parameters against a set of model organisms (Human, Zebrafish, Mouse, Drosophila) with well annotated transcription factor

databases (following AnimalTFDB v3.0)[104]. For the latter, a given *Pristina* gene was counted as TF if at least another TF gene from any of the species belonged to the same orthogroup as the *Pristina* gene. The different sources of evidence were pooled together and we kept those *Pristina* genes with at least two independent sources of TF evidence. Every TF gene was assigned a class based on their sources of evidence.

### Transcription factor analysis

The CPM table was subset to retrieve the *Pristina* TFs, and gene expression across cell types was scaled and visualised using the ComplexHeatmap package[105]. To analyse the TFs at the class level, for a given class X, we calculated the median and average coefficient of variation (CV) of class X across cell types, the number of genes pertaining to class X, and the cumulative number, average, and median counts of class X. We visualised the relationship between CV and number of genes using the base and ggplot2 packages (https://ggplot2.tidyverse.org/) in R v4.0.3 (https://www.R-project.org/).

We did a multivariate analysis two-way ANOVA to detect differences in TF expression between cell clusters, TF classes, and the interaction of the two. TF counts were aggregated at the broad cell cluster level and we kept only those TFs from classes with four or more annotated genes. The ANOVA was run using aov(), followed by Tukey comparison of means using TukeyHSD(). The most prominent classes explaining differences across cell clusters were retrieved by quantifying and sorting the results of the Tukey test.

To represent these differences visually, we calculated the expression prominence of each TF class (the sum of counts per gene). For a given TF class X, we defined the prominence of class X across cell clusters as the addition of the counts of all genes of class X in each cluster, divided by the number of genes of class X expressed at each cluster. The resulting matrix was normalised and visualised using a custom ggplot2 wrapper function in R v4.0.3.

### WGCNA analysis

We ran WGCNA[68] using a subset of the CPM table to keep genes with CV > 1 and softPower 5 estimated after visualising the Scale-Free Topology Model Fit. Adjacency and Topological Overlapped (TOM) matrices were calculated using standard parameters. For dynamic cutting of the tree, we chose 100 genes as minimum module size. Provided the discrete expression of gene modules, these were named and recolored manually following a similar criterion than when naming cell clusters. The resulting classification in modules was used to reorder the expression dataset, and the dataset was represented for visualisation using ComplexHeatmap[105].

To calculate the association between TF classes and modules, we calculated the connectivity of each TF gene to each module eigengene. For a given TF class X, we quantified the number of genes of class X with a connectivity equal or higher than 0.5 to each module eigengene. The resulting matrix was normalised and represented using the package ComplexHeatmap.

WGCNA graphs were constructed using the TOM matrix and pruning from sparse interactions using an arbitrary low threshold of connectedness (>0.01). A subset of the resulting graph (>0.35) (hereafter "0.35 graph") was used for exploratory analysis using the igraph package[106] and the Fruchterman-Reingold layout algorithm[107] with parameters 'maxiter = 100 * NUM_GENES_GRAPH, kkconst = NUM_GENES_GRAPH', where NUM_GENES_GRAPH is the number of genes present in the 0.35 graph. Connected component membership was calculated using the function components() from the igraph package, and its percent of agreement with the WGCNA module membership was calculated using the adjusted Rand Index implementation adjustedRandIndex() from the package mclust[108]. The 0.35 graph was subdivided into subgraphs corresponding to the connected components using a custom wrapper function that implements the induced_subgraph() function of the igraph package. Centrality of the TFs belonging to each separate sub-graph was calculated using the closeness() function from the igraph package in a custom wrapper function, and visualised using ggplot2.

We used a less stringent subset of the 0.01 graph (>0.2, rather than 0.35) to analyse cross-module connections. Using a custom wrapper function, a 'gene x module' matrix was constructed counting how many genes from each module are direct neighbours to a given gene x, and normalised by dividing the number of connections of gene x to each module by the size of the module that gene x is part of. These numbers were later aggregated at the module level to retrieve the number of normalised cross-connections between modules. The resulting matrix was transformed into a graph using graph_from_adjacency_matrix() from igraph with parameters 'mode = "upper", weighted = TRUE, diag = FALSE', and the number of cross-connections was used for edge size to highlight the largest amounts of cross-connections.

### Limma analysis

Differential Gene Expression Analysis was performed using the edgeR[109] and limma[110] R packages, and the pseudo bulk UMI count matrix. Briefly, we made a distinction between 'piwi-positive' and 'piwi-negative' cell clusters in order to retrieve the genes that are differentially expressed in 'piwi-positive' cells. A DGE object was created using the counts table and a sample information table with the aforementioned distinction, as well as a model matrix. The dataset was filtered using the filterByExpr() function from edgeR, and normalised using the voom() method from limma. Linear modelling was done using the lmFit() function with the model matrix (all 'piwi-positive' vs 'piwi-negative'), and statistics were calculated with the eBayes() function. The results were plotted using the EnhancedVolcano (https://github.com/kevinblighe/EnhancedVolcano) and ggplot2 packages.

### Gene Ontology analysis

Gene Ontology (GO) analyses were performed using the R package topGO[111] and the 'elim' method using a custom wrapper function. GO terms with less than three significantly annotated genes were discarded. Unless otherwise specified, we chose the totality of *Pristina* genes as the gene universe population to compare against.

### *Piwi*+ cell transcription factor analysis

For this analysis we used raw UMI counts extracted at the broad cell type group and normalised them as described above. Then, the relative enrichment of expression in each broad cell type group was calculated by subtracting the log cpm (with a pseudocount) of each cell type from the mean log cpm (with a pseudocount) of the remaining broad types. We then filtered this table to contain only TFs and extracted those with the higher coefficients of variation (cv >1). We used this table to sort the top 200 TFs with higher levels of enrichment (log ratios) in *piwi*+ cells compared to all other cell types (Supplementary Data 11).

### Epigenetic factor analysis

We extracted lists of epigenetic factor components from https://epifactors.autosome.org/[78,79], containing human protein sequences. We then blasted those against the translated *Pristina* transcriptome using tblastn. We manually curated the selection of top hits for each epigenetic factor, and annotated those that are annotated as members of more than one epigenetic regulation complex (Supplementary Data 14).

### in situ HCR hybridisation

For in situ Hybridisation Chain Reaction (HCR), previously published protocols[112] were used with mainly modifications for *Pristina leidyi*

fixation and 1st day of the protocol, based on the species colorimetric in situ hybridisation protocols[5]. Specifically, samples were fixed in 4% PFA for 40-45 minutes, dehydration/rehydration steps in methanol were skipped, and after washes in 1x PBSt, in situ HCR protocol was carried out on the same day. Day 1 of the original colorimetric in situ hybridisation protocol (which includes pronase digestion, acetylation, and post-fixation) was found to be essential for successful results in *Pristina leidyi*. The entire protocol can be accessed in https://github.com/BDuyguOzpolat/Pristina_leidyi-protocols.

EdU labelling of proliferating cells was incorporated into the in situ HCR protocol with minor modifications, following the SHInE protocol[113]. A 0.5 mM EdU solution in 1% filtered artificial seawater was prepared from a stock solution of 100 mM EdU in DMSO. Worms were incubated inthe EdU solution for 24 h before fixation. The Click-it reaction was performed with 5 μM Alexa Fluor™ 568 dye between the hybridisation and amplification steps.

**Selection of markers and designing probesets.** For each cell cluster, top expression markers with coding sequence length of 700 bp or longer were listed (for compatibility with HCR probe design). Probesets were designed for 1 or 2 of these markers per cluster using the Özpolat Lab algorithm (https://github.com/rwnull/insitu_probe_generator)[112]. The sequences used for probe design were confirmed to be in 5′ to 3′ orientation using https://web.expasy.org/translate. For each probeset, the lower probe pair limit was 11 and the upper limit was 34 pairs. Complete list and sequences of probesets, along with the associated initiator information can be found in Supplementary Data 6.

Buffers and hairpin amplifiers were ordered from Molecular Instruments[114]. For all in situ HCR experiments a combination of the following hairpin-fluorophore conjugations were used: B1-546, B2-488, B3-647, B4-594, B3-594, B4-647, B4-488.

### Confocal imaging
Confocal imaging was carried out using Zeiss LSM710 and LSM780 microscopes at the microscopy facility at Marine Biological Laboratory, and a Zeiss LSM800 microscope at the Oxford Brookes Centre for Bioimaging. For each set of HCRs, control tubes were included. Controls did not have any probes, but had hairpins, in order to assess the unspecific background signal (Supplementary Fig. 6). Image analyses and editing were carried out in Fiji[115], panels and schematics were prepared using Adobe Illustrator. Stiching of the tiles was done using the Fiji "Pairwise stitching" plugin[116]. Either single plans or maximum projections of z-stacks were chosen for the figures.

### Nuclei area quantification
For comparison of *vigilin*+ cell nuclei size with the other cell types in the area, we used the nuclear staining in confocal Z-stacks, and measured the area for each nucleus using Fiji[115]. 3 different worm samples were used for measurements. Samples were imaged as z-stacks, and the nuclei to be measured were picked from 5 focal planes across the stack. At each focal plane 5 nuclei for *vigilin*+ cells and 5 nuclei from the nearby cells that are negative for *vigilin* were measured (25 nuclei each group, 50 nuclei per sample). The R Wilcoxon rank sum test (wilcox.test) was used for statistical analyses using R to compare the two groups.

### Subclustering *piwi*+ clusters
We selected *piwi*+ cells by selecting cells in clusters 1, 2 and 8, including 16,247 cells, and we reanalysed them alone from the raw unprocessed matrix. We calculated metrics using sc.pp.calculate_qc_metrics, and normalised the matrix using sc.pp.normalize_total with a target_sum=1e4. We selected high variable genes using sc.pp.highly_variable_genes with n_top_genes = 18000, and sliced the matrix to contain only those genes, storing the raw in an adata.raw object. We then

scaled the matrix with sc.pp.scale, performed pca with sc.tl.pca, constructed a kNN graph with sc.pp.neighbours, with 35 neighbours and 25 principal components, and calculated a UMAP visualisation with sc.tl.umap (min_dist=0.5, spread = 1, alpha = 1, gamma = 1.0). We run the Leiden clustering algorithm using sc.tl.leiden with resolutions 0.4, which gives 10 clusters. We calculated marker genes for each cluster using sc.tl.rank_genes_groups using both the Wilcoxon (method = 'wilcoxon') and the Logistic Regression (method = 'logreg').

### Scores
To calculate gene scores we used the Scanpy function sc.tl.score_genes with a control size equal to the length of the gene list and a number of bins equal to 25.

### Reporting summary
Further information on research design is available in the Nature Portfolio Reporting Summary linked to this article.

## Data availability
The sc-RNA-seq reads and the cell matrix generated in this study have been deposited in the GEO database under accession code GSE230505 and are also listed in Bioproject PRJNA961657. The Iso-seq reads generated in this study have been deposited in the BioSample database under accession code SAMN34360745 [https://www.ncbi.nlm.nih.gov/geo/query/acc.cgi?acc=GSM7225503]. The sequence and annotation references used in this study are available in the following databases: BUSCO, nr [https://www.ncbi.nlm.nih.gov/refseq/about/nonredundantproteins/], Pfam [http://pfam-legacy.xfam.org/], PANTHER, SUPERFAMILY, AnimalTFDB v3.0 [https://guolab.wchscu.cn/AnimalTFDB], SwissProt [https://www.uniprot.org/], EggNOG [http://eggnog5.embl.de] and EpiFactors [https://epifactors.autosome.org/].

## Code availability
The code used for all the analyses in this study is available in GitHub (https://github.com/scbe-lab/pristina-cell-type-atlas) as well as Zenodo (https://doi.org/10.5281/zenodo.10671442)[117].

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

## Acknowledgements

The authors thank Robert Hedley and Vasiliki Tsioligka at the Flow Cytometry Facility at the Dunn School of Pathology (University of Oxford), the MBL Imaging Facility, and Ryan Null with in situ HCR probe design assistance. We thank Maria Rossello for discussions about the transcriptional landscape analysis and the DGE analysis. Research at the Solana lab at Oxford Brookes University is supported by MRC grants (MR/S007849/1 and MR/W017539/1), a Royal Society Grant (RGS\R1\191278), a BBSRC Grant (BB/V014447/1) and a Leverhulme Trust grant (RPG-2019-332) to JS. Research at the Álvarez-Campos lab was supported by the European Molecular Biology Organization funding (EMBO Long Term Fellowship to PA-C, ALTF-217-2018) and the Comunidad de Madrid-Spain Government (Regional Program of Research and Technological Innovation, SI1/PJI/2019-00532). Research at the Özpolat lab is supported by NSF (1923429-EDGE CT), NIGMS (1R35GM138008-01) grants and Hibbitt and WashU Startup Funds. The generation of the *Pristina leidyi* transcriptome and the initial single cell atlas experiments were supported by two Research Excellence Awards from Oxford Brookes University to NJK and JS respectively. HG-C and EE were supported by Nigel Groome studentships from Oxford Brookes University. Two Travelling Fellowships from The Company of Biologists supported HG-C's visit to the Özpolat laboratory (DEVTF2108578) and IdO to the Solana laboratory (DEVTF2110590).

## Author contributions

P.A.-C., H.G.-C., J.S. and B.D.O. conceived the study and designed the experiments. P.A.-C., H.G.-C. and E.E. generated cell dissociations and performed single-cell transcriptomic experiments using *Pristina leidyi*, assisted by V.M. H.G.-C., B.M., I.d.O., S.P. and B.D.O. generated in situ HCR data. N.J.K. performed bioinformatic experiments on the *Pristina leidyi* transcriptome and initial bioinformatic single-cell analyses. A.P.-P. performed bioinformatic analyses on the transcriptional landscape of *Pristina leidyi*. D.A.S.-D. performed bioinformatic single-cell analyses. J.S. performed bioinformatic single-cell analyses and *Pristina leidyi* piwi+ population transcriptomic analyses. A.E.B. contributed to the interpretation of the single-cell analysis data. J.S., B.D.O., P.A.-C. and H.G.-C. wrote the manuscript and generated the figures, with contributions from all other authors. All authors read and approved the final version of the manuscript.

## Competing interests

The authors declare no competing interests.
