## [Peer Review File · Nature Communications]

Annelid adult cell type diversity and their pluripotent cellular originsREVIEWER COMMENTS

Reviewer #1 (Remarks to the Author):

In "Annelid adult cell type diversity and their pluripotent origins", Alvarez-Campos and co-workers use an impressive dataset of 75,218 single-cell transcriptomes to report the most detailed characterisation of the cell type diversity of an adult annelid to date. Annelids are a species-rich animal clade of economic and ecological importance that play a central role in developmental, regenerative and evolutionary biology. However, our understanding of their anatomy, morphology, and underlying genetic features is limited, impacting our ability to answer fundamental questions, such as the cellular and molecular basis for the extensive regenerative ability observed in some annelid species. To address this knowledge gap, the authors use single-cell transcriptomic profiling with the SPLIT-seq approach on the small asexual and highly regenerative adults of the species *Pristina leidyi*. Data analysis recovered 60 cell clusters corresponding to 17 large cell types (each with sub-clusters) that were experimentally validated with in situ HCR. Some of these cell types likely correspond to either novel or poorly understood ones and reveal unexpected biology of these animals, such as the anterior-posterior regionalisation of the digestive tract. The two largest clusters, however, correspond to piwi+ cells and display molecular features often associated with stem cells in other systems. Importantly, some of these piwi+ cells co-express transcription factors that are markers of other differentiated cell types, which, together with lineage trajectory analysis, suggest that the regenerative capacity of *Pristina* might rely on a general population of pluripotent stem cells that commit and differentiate into different cell lineages, as observed in other invertebrates with strong full-body regenerative abilities such as planarians and acoel worms. Overall, this solid and thorough work provides a valuable resource for the large developmental and regenerative research community, setting numerous working hypotheses for future investigation in this and other annelids.

I find, however, some points that might need further clarification and/or additional evidence:

- The authors apply three independent approaches to remove doublets (Scrublet, Solo and manual threshold) and still, some of the retained clusters remain as potential doublets and are deemed unannotated. These clusters largely overlap with piwi+ cells. Although they represent a small fraction of the dataset, could they have influenced the analysis of TF co-expression in piwi+ cells? This is a key finding of the manuscript that suggests the existence of lineage-committed piwi+ stem cells. Surprisingly, it is also not experimentally validated (e.g., by doing HCR of piwi and the TFs, at least for some major trajectories). This would provide a more detailed characterisation of these potential stem cells and transitory states, strengthening the central messages of the study (e.g., do cells co-expressing some TFs and piwi are closer to the differentiated cell type? Do these cells occur mostly in regenerating individuals? Etc.). The restriction of epigenetic regulators to piwi+ cells also sounds strange. Epigenetic regulation likely occurs in all cells. Is the expression of a certain set of epigenetic regulators, and not others, what is remarkable?

- Vigilin+ cells, salivary glands and polyploidy. Nuclear size can also correlate with transcription activity without that implying polyploidy (e.g. <https://pubmed.ncbi.nlm.nih.gov/7523596/>), and as the authors acknowledge, these cells have high RNA counts per cell. Is it possible to experimentally validate the presence of multiple chromosomes (e.g. using colchicine) in these cells? That will solidify their interpretation. Regarding the evolutionary scenario (lines 356-358), salivary glands with polyploid cells appear largely restricted to some insect groups. Therefore, if demonstrated, polyploidy in this cell type would likely be convergent between annelids and insects and not a case of deep homology.

- While the dataset and findings are solid and high quality and largely justify the conclusions, the text lacks context and a broader picture (especially in the Intro and Discussion) and is sometimes repetitive. For example, while the abstract reads more like a series of findings, the last paragraph of the Introduction is more engaging and better addresses the overarching, and perhaps more generally

appealing, question of the manuscript. Likewise, the Introduction focuses on the technical value of scRNA-seq, but it could provide a better picture of the knowledge gaps in stem cell biology and regeneration in annelids (what is known and needs to be solved). Some of the text in the Results section is redundant or could fit better in Methods, especially in the first section (e.g., sequencing mode, etc), and I wonder if Supplementary Note 1 could be incorporated within the main text, blended with some of the text from lines 152 to 236 under a broader section on cell type diversity. Indeed, it is unclear why the validation of epidermal, muscular and neuronal cell identities is a separate section when all main clusters are generally experimentally validated. Some aspects of the Results are also poorly explained. For example, in lines 262-264, when referring to the centrality of TFs in co-expression networks. What does that mean? What is the biological insight that derives from this analysis? Finally, the Discussion acts more as a Conclusion section, summarising and repeating the main findings rather than putting them in a broader context. For example, the end of the sentence in line 233 gives more or less the same idea as the following sentence (and see point above regarding polyploidisation) and lines 361 and 362 of the Discussion.

Other points:

- Line 35: "cell type diversity of adult annelids for the first time"; by scRNA-seq approaches because many other methods have been used to characterise this animal group's anatomical and cellular diversity.
- Lines 94-96: how many of the transcripts have an annotation?
- The authors use the terms counts and UMI interchangeably, but it might be better to be consistent.
- Line 132: Figure 2E, missing closing bracket.
- Line 152: HCR in situ hybridisation should be in situ HCR.
- Line 165: the longitudinal fibres are not obvious from the HCR pattern; a phalloidin staining on a separate sample may help.
- Line 191: glial-like cells have been described in annelids (e.g., <https://royalsocietypublishing.org/doi/10.1098/rspb.2017.0743>), so their existence as a separate cell type is not new.
- Line 197: "We also found annelid types" should be "We also found clusters"?
- Line 255: eleocyte are mentioned here for the first time without proper description.
- Line 340: "species capable of extensive adult cell type generation and regeneration" is an odd expression. I guess the authors mean *Pristina* has a diverse repertoire of cell types that can be regenerated?
- Line 334: "this shows that epigenetic regulation is a conserved feature of animal pluripotent cells." Is this surprising? Epigenetic regulation occurs in all cells and is
- In figures and panels of HCR, wouldn't be better to refer to the gene being used for HCR with the standard orthology instead of the contig name? If there is an annotation, of course. It makes it easier and more informative for the reader.

Reviewer #2 (Remarks to the Author):

Alvarez-Campos et al. sought to describe the cell type composition and diversity in the annelid *Pristina leidyi* (referred to as *Pristina* further on) by means of computational analysis of a single-cell transcriptional atlas. Taking this approach, they characterized and validated the major cell types in *Pristina*, which includes gut, muscle, epidermal and neuronal cells. In addition, they also identified and validated annelid-specific cell types, some of which have not been previously reported (*Idlrr+* cells and carbohydrate metabolic cells). With the power of this cell type atlas, they also sought to investigate the putative stem cells in *Pristina*. *Piwi+* cells have been studied in *Pristina* in the context of the germ line, but given the role of *piwi+* stem cells in highly regenerative animals, they sought to investigate whether the *Pristina piwi+* cells share characteristics with potentially homologous cell types in other

systems. Altogether, they present a characterization of the cell type diversity in *Pristina* and provide molecular data that suggest a stem cell-like role to the population of *piwi+* cells in *Pristina*.

This study is valuable as a resource as it provides a molecular characterization of the cell types in *Pristina*. Cell types atlases at this resolution have proven to be useful in many other invertebrates (planarians, sponges, cnidarians, acoels, etc), given its power to both validate their cell type diversity and open the ground for new hypotheses to be tested functionally. While the characterization of the major cell types in *Pristina* was validated through HCR in situ hybridization and confirms their hypothesized expression patterns, the study lacks functional characterization of the putative stem cell population in *Pristina*. Stem cells in the previously-studied invertebrate systems have been characterized by means of molecular and functional analysis— the statements about the distribution, potency and differentiation dynamics of *Pristina piwi+* cells remains to be validated using functional tools (HCR, IHC, EdU, etc). The single-cell RNA-seq data present an exciting hypothesis of these cells being stem cells akin to neoblasts or *i*-cells; however, molecular trajectories are not evidence of this property, they are hypotheses that need to be validated. This work is high-quality and will have high impact on the field, but it is very important for the authors to restate their claims to make clear that many of them relating to the *piwi+* population are hypotheses. It would also be important to clearly outline alternative hypotheses that could explain the data. This does not weaken the paper but instead strengthens it; it lays out a roadmap for important work to be done next. See below for details on these major comments.

1) Issues with the UMAP and PAGA plots in Figure 1

The current clustering of cells in Figure 1B, while being clear in sense of the chosen color-code, lacks organization in the annotation of some clusters. In some instances, it is complicated to determine 1) which clusters are annotated and which ones are not, and 2) what cluster numbers belong to which clusters. Here are two examples:

- Overlap between cluster numbers [such as 47 overlapping a number below it] clouds the assignment of these cluster numbers to their respective clusters.
- The presence of more than one cluster number too close to each other [such as 45 and 2 on the same gray cluster], making it hard to identify what clusters those numbers belong to.

This becomes an issue when trying to understand the PAGA graph in Figure 1D. Given that the PAGA algorithm reconstructs lineages by means of the topology of a given UMAP, it is hard to visually assess the vicinity of a cluster to its connected clusters. For instance, cluster 2 connects with clusters 21, 34 and 41, but it is complicated to see where those clusters are in Figure 1B. Overall, organizing labels and annotating the clusters properly will allow for a streamlined understanding and takeaway of the UMAP.

2) Functional validation of *piwi+* cells in *Pristina* is lacking

While *Pristina piwi+* cells have been explored in the context of gonadal dynamics in Ozpolat & Bely 2015 and in Ozpolat et al 2016, the present study lacks a clear definition and validation of the *Pristina piwi+* cells in terms of their putative identity as stem cells. Figure 5A suggests that these *piwi+* cells are proliferative, based on the expression of proliferation markers (*pcna*, *h2b*, *h2a*) in these *piwi+* cells; validation of this could be provided by developing a double *piwi* HCR in situ with the labeling of the mitosis marker H3P. This experiment would not only confirm these cells to be proliferative, but also provide information about the proportion of *piwi+* cells that are proliferative and the distribution of these cells along the *Pristina* body plan. Further, if these cells are proliferative, the EdU protocol published in Ozpolat et al 2016 could be used to trace EdU+ cells (which would presumably also be *piwi+* cells) and ask whether they differentiate into multiple cell types. These experiments would allow to explore the cellular properties and differentiation potency of the *Pristina piwi+* cells. The goal of mentioning these experiments is not to suggest that the authors must do them all for the context of this manuscript, but to underscore that claims about the properties of this cell type, if it is one cell type at all, need to be supported with experimental evidence of its existence, its distribution, and its proliferation and differentiation properties. Without that, all statements need to be tempered and presented as hypotheses. Key things that need to be added/modified:

- An explanation of whether HCR was attempted on piwi and all the other markers of the piwi+ population found in this study.
- Given that there is no corroboration provided on which cells express these markers and where they are located in the Pristina body plan, it is important to consider that Pristina might be more similar to vertebrates in that there may be distinct tissue-restricted specialized stem cell populations that are unified in their expression of proliferation genes, etc. This is a very different scenario from them being pluripotent stem cells like neoblasts or i-cells. This scenario would look indistinguishable from the pluripotent population scenario in terms of single-cell RNA-seq. Therefore, the authors should include this and any other alternatives as they interpret their data.

3) Overinterpretation of PAGA results

The authors rely heavily on their graph abstraction algorithm to make conclusions about the putative stem cell population. PAGA is very useful for quantifying interconnectedness of clusters, but it is important to remember that ultimately that interconnectedness is only based on gene expression similarity. Gene expression similarity can be suggestive of a differentiation trajectory, but it cannot serve as evidence of differentiation. It is interesting to note that several small clusters were recovered close to the putative stem cell clusters, and therefore it is no surprise that PAGA then shows the stem cell cluster as being connected to more other clusters than any other cluster. This is not at all convincing as evidence of potency. See e.g. Lines 289-290: "The highest potency score in our abstracted graph was attained by piwi+ cell cluster (Figure 5D), strongly suggesting that piwi+ cells are pluripotent stem cells."

4) Statements and inferences are not supported by the data presented

Along the results and discussion sections, particularly in relation to the Pristina piwi+ cells, the authors' conclusions and inferences based on the data presented are often not consistent. Providing data supporting the hypothesis that one cell population is pluripotent, and thus able to give rise to all cell types in Pristina, requires in vivo lineage tracing, which was not presented in this study. Therefore, statements regarding the pluripotent identity of cells in Pristina should be edited to convey what the data really presents. Below are some sentences that should be edited (this list is not exhaustive and authors should revise other related statements):

- Lines 281-283 "Moreover, our PAGA analysis revealed that most differentiated cell types were connected to piwi+ cells by differentiation trajectories (Figure 1D, Figure 5B), including epidermis, muscle and gut, suggesting these cells are a pluripotent population." Once again, lines drawn by PAGA are really only showing quantified interconnectedness in terms of gene expression similarity which can serve as hypotheses of differentiation. The lines should not be interpreted as de facto differentiation trajectories. Yes, PAGA incorporates RNA velocity, but there is simply no way to know in an unvalidated dataset how effectively that is capturing differentiation.
- Lines 289-290: "The highest potency score in our abstracted graph was attained by piwi+ cell cluster (Figure 5D), strongly suggesting that piwi+ cells are pluripotent stem cells."
- Lines 303-304: "Altogether, these analyses showed that piwi+ cells in Pristina are a heterogeneous cell population with a pluripotent stem cell signature."

Lines 334-337: "Taken together, our data strongly suggest a model where post-transcriptional and epigenetic regulators control stem cell maintenance and pluripotency, and a panoply of TFs prime these to differentiate into multiple cell types."

Minor comments

- Lines 56-58: Provide reference for other single-cell atlases: from planarians (Fincher et al 2018) and acoels (Duruz et al 2021, Hulett et al 2023)
- Lines 75-77: Sentence feels incomplete
- Lines 142-143: It would be helpful to specify how the PAGA algorithm works and why in this case is more helpful to use PAGA over other algorithms to reconstruct the differentiation trajectories.
- Line 265: Centrality of a TF could also suggest their importance in the functionality of the mature cell types, not just only their differentiation.
- Line 300-301: Specify what subcluster this is

- Figure 1D and Supplemental Figure 4: It is unclear why there are some clusters that appear in Supplemental Figure 4 but not in Figure 1D (for example, cluster 48, 57 and 58 are in Supp4 but not in 1D)
- Figures 3B, 3C, 3H, 3G, 3I: Are the extensions in the outside of the body plan also positive for their assay, or is the staining considered background?
- Figure 5C: Specify the names of the transcripts in the dot plot
- Supplement Figure 9: Specify the transcript names of each of the UMAP projections

Reviewer #3 (Remarks to the Author):

The work under consideration represents a convincing effort to understand the cellular landscape in an asexually-reproducing and regenerating clitellate annelid, the oligochaete *Pristina leidyi*. A total of 75k cells have been transcriptome-sequenced with the SPLiT-seq method. The authors are to be commended for producing a technically sound and interesting dataset that will serve the *Pristina* and annelid community to advance on the expression characterization of stem cells and differentiated cell types in this and other annelids. In its current state, however, the analysis remains rather shallow and does not go beyond a cursory identification of markers for a subset of the identified cell clusters, mostly at a rather low resolution such as 'epidermis', 'muscles', or 'neurons'. For the gut, the authors go into more detail and reveal a striking anterior-posterior regionalization of the gut into different cell populations with sharp boundaries that are nicely documented by HCR in situ hybridization. Most emphasis is on the presumed stem cell fraction of the single cell dataset, which is analyzed via partition-based graph abstraction (PAGA) of the connectivity between cells. This identifies a population of cells connecting to many possible differentiation trajectories and thus most likely representing pluripotent stem cells. This is corroborated by the expression of proliferation and chromatin regulators in these cells. Overall, this is an important addition to the growing number of single-cell datasets in different animal phyla, and will represent an important comparative resource for the regeneration and evolutionary cell biology communities. While it falls short of the in-depth analysis and identification of cell types, it opens up a new level of understanding of pluripotent progenitor and stem cell populations in an important new model for asexual reproduction and regeneration, complementing the study of planarians.

Some points for improvement in order of decreasing relevance.

- It is not clear why there is not much of a comparison to the Schmidtea data and how the different datasets compare; what is the relationship to Schmidtea neoblasts, their potency and differentiation potential, their transcription factor and regulator signature etc. These data are available and a comparison is straightforward using SAMap, data integration approaches or alike. Schmidtea shares with *Pristina* the major characteristics relevant here, i.e. asexual reproduction and high regenerative potential.
- The study is technically high-quality, including for example a new, rather complete long-read reference transcriptome, stringent filtering, and high cell numbers that help overcome the low numbers of UMIs/cell. It is less clear what exactly has motivated the clustering cut-off but the clusters defined by the authors appear sufficiently differentiated for this to not be critical. However, the homogeneity of the clusters presented in Figure 1B would be more accessible if instead of feature plots as in Fig. 1C, a heat map would be shown for the 50 most specific genes for each cluster and their expression across all cell types (an extended version of the dotplot in Suppl Fig. 1E). Since the authors were more interested in (re)generation, the in-depth annotation of the cell atlas is of lesser importance.
- For the transcription factor analysis in Fig. 4, the authors' approach to identifying TFs is valuable and much appreciated. It is a pity though that the identity of the TFs defining the clusters cannot be accessed. Fig. 4C should be presented as a "staircase" plot as in 4A and TF names be added (e.g., in a

Suppl. Fig.). The TFs and the table underlying Fig. 4C should be made available. As above, the feature plots presented in Fig. 4D are the least favorable solution to illustrate TFs versus clusters and could go to supplementary material, replaced by dotplots. How many cells express the TF per cluster?

- The graph analysis is confusing. WGCNA by design groups genes that show correlated expression; it should be no surprise that clustering a graph made from the TOM matrices, which are needed to cluster the genes into modules in WGCNA, shows the same thing (“[...] identified several connected components that reliably match the WGCNA modules”). Furthermore, not having seen this type of analysis elsewhere, is there is a more straightforward way to arrive at a similar conclusion, e.g. by just calculating correlation between all WGCNA genes and placing cutoffs on that to form a graph?

- The network visualization of the WGCNA modules shown in Fig. 4E appears to produce a measure of similarity between cell types by means of shared gene co-expression. How is this going beyond what is already shown in Fig. 4A? The stomach-neurons-secretory-intestine-cluster in the 5 o’clock portion of 4E is surprising, as Fig. 4A does not seem to indicate much co-expression between at least most of these modules. What is the significance?

- The authors mention that they used arbitrary cut-offs to build the WGCNA graph. While many score distributions follow power laws, sometimes the underlying distribution is bimodal. Did the authors investigate that?

- The claim that vigilin+ cells are polyploid cells that function as glands is very speculative and should be toned down.

- The PIWI analysis is in the heart of the study. Given that, it will benefit from some additional detail. Did the authors try to expand the PAGA analysis to the subclustered piwi+ cells (all 8 subclusters from the supplementary figure)? This might offer some clarity that the UMAP cannot provide, also with regard to the possible pluripotency of cells. Also, the identity of the TFs expressed in Piwi+ cells should be revealed and how they relate to the subclusters.

- Polycystin not “cistin”

Reviewer 1

Reviewer #1 (Remarks to the Author):

In "Annelid adult cell type diversity and their pluripotent origins", Alvarez-Campos and co-workers use an impressive dataset of 75,218 single-cell transcriptomes to report the most detailed characterisation of the cell type diversity of an adult annelid to date. Annelids are a species-rich animal clade of economic and ecological importance that play a central role in developmental, regenerative and evolutionary biology. However, our understanding of their anatomy, morphology, and underlying genetic features is limited, impacting our ability to answer fundamental questions, such as the cellular and molecular basis for the extensive regenerative ability observed in some annelid species. To address this knowledge gap, the authors use single-cell transcriptomic profiling with the SPLIT-seq approach on the small asexual and highly regenerative adults of the species *Pristina leidyi*. Data analysis recovered 60 cell clusters corresponding to 17 large cell types (each with sub-clusters) that were experimentally validated with in situ HCR. Some of these cell types likely correspond to either novel or poorly understood ones and reveal unexpected biology of these animals, such as the anterior-posterior regionalisation of the digestive tract. The two largest clusters, however, correspond to *piwi*+ cells and display molecular features often associated with stem cells in other systems. Importantly, some of these *piwi*+ cells co-express transcription factors that are markers of other differentiated cell types, which, together with lineage trajectory analysis, suggest that the regenerative capacity of *Pristina* might rely on a general population of pluripotent stem cells that commit and differentiate into different cell lineages, as observed in other invertebrates with strong full-body regenerative abilities such as planarians and acoele worms. Overall, this solid and thorough work provides a valuable resource for the large developmental and regenerative research community, setting numerous working hypotheses for future investigation in this and other annelids.

We would like to thank the reviewer for highlighting the importance of our work, the excellence of the dataset we provided, and the solidness and thoroughness of our work.

I find, however, some points that might need further clarification and/or additional evidence:

- The authors apply three independent approaches to remove doublets (Scrublet, Solo and manual threshold) and still, some of the retained clusters remain as potential doublets and are deemed unannotated. These clusters largely overlap with *piwi*+ cells. Although they represent a small fraction of the dataset, could they have influenced the analysis of TF co-expression in *piwi*+ cells? This is a key finding of the manuscript that suggests the existence of lineage-committed *piwi*+ stem cells.

We thank the reviewer for this interesting comment. Indeed, we performed a very careful analysis of the doublets because we share this concern. Doublets are an inherent property of single cell transcriptomics, both by combinatorial barcoding and by droplet methods (where other concerns such as ambient RNA are also important). Doublet detection methods are typically trained/optimised in datasets with a lower complexity than our whole organism atlas. These methods typically work by generating artificial doublets to see how they would look like and then searching for similar cells in the dataset. However, when the complexity is high (over 50 distinct clusters, with frequencies that range from as high as 10-20% to as low as 0.3% of the dataset), this doublet detection task is also very complex. Indeed, a doublet of a differentiated cell and a *piwi*⁺ cell would generate a cell barcode that would have both *piwi* and other stem cell related genes and differentiated genes as the reviewer points out, including TFs. If the “unannotated” clusters are doublets, indeed they could affect the analysis of TFs in the *piwi*⁺ cell population.

We think that the nature of these unannotated cells could be doublets but also they could be true *piwi*⁺ cells that are being artifactually clustered out. Clustering algorithms normally handle poorly the presence of very lowly abundant and very highly abundant cell populations in the same sample. Therefore, to cluster out very rare populations, some overclustering of the most abundant types normally happens.

We can offer the following arguments to reassure the reviewer that the unannotated cells, interpreted as doublets, are not the source of the TF heterogeneity observed, or, at the very least, not the major driver.

- 1) If doublets *piwi*⁺ cell/differentiated cell are happening, we can expect a similar rate of differentiated cell/differentiated cell doublets, at least for the major ones. These should create artefactual differentiation trajectories between two differentiated cell types and TF (and other differentiated marker expression) of each one of the types within the other. We are not seeing this as a major effect. This indicates that the differentiation trajectories that we are seeing, which are inherently connected with the observation of *piwi*⁺ cell heterogeneity, are of biological origin.
- 2) The unannotated clusters do overlap with the *piwi*⁺ cell population, but were considered not part of the *piwi*⁺ cell population in the heterogeneity analysis. This means that further similar doublets should be still present (other than those detected by the 3 layers of doublet removal that we applied) in order to affect this analysis. We performed a very stringent analysis, where we aimed at removing doublets as a possible source of heterogeneity by 3 different methods to make this analysis as solid as possible.

3) While the unannotated clusters do express some TFs, there are plenty of other TFs where we observed expression in the *piwi*+ cells but not in the unannotated clusters. This can be observed in the dotplot, where, for instance, PrileiEVm10521t1 is a TF expressed in gut and *piwi*+ cells. In fact, other TFs are expressed in the unannotated fraction prominently: we believe that this is because most likely, these unannotated cells that overlap with the *piwi*+ cell population are indeed *piwi*+ cells that are clustered out by the clustering algorithm as explained above. Under this interpretation, even after removing the most heterogeneous cells in the *piwi*+ cell population, we still observed substantial heterogeneity, reinforcing rather than refuting this same notion.

Surprisingly, it is also not experimentally validated (e.g., by doing HCR of *piwi* and the TFs, at least for some major trajectories). This would provide a more detailed characterisation of these potential stem cells and transitory states, strengthening the central messages of the study (e.g., do cells co-expressing some TFs and *piwi* are closer to the differentiated cell type? Do these cells occur mostly in regenerating individuals? Etc.). The restriction of epigenetic regulators to *piwi*+ cells also sounds strange.

We have attempted experimental validation of these observations, aiming at observing co-expression of TFs with *piwi*. Unfortunately, these probes gave low signal, likely because of the relatively low expression levels of both *piwi* and the TFs. However, in the new version of the paper we have added a new figure (new Figure 7) with extra validations which hopefully will reassure the reviewer. In these validations, we picked markers of *piwi*⁺ cells (that work better as HCR probes) and a selection of differentiated cell markers with detected expression in the *piwi*⁺ cells (that similarly work better as HCR probes) and managed to colocalise them all with *piwi*⁺ cells, validating the expression of differentiated cell markers within the *piwi*⁺ cell population. While this is not exactly what the reviewer asked it is parallel evidence that likely is underlied by the same principle: annelid *piwi*⁺ cells express markers of several differentiated types, likely implying that they have the capacity to differentiate into them. Future studies will attempt the colocalisation of TFs and *piwi*.

Epigenetic regulation likely occurs in all cells. Is the expression of a certain set of epigenetic regulators, and not others, what is remarkable?

Epigenetic regulation likely occurs in all cells indeed, but the expression of epigenetic regulators tends to be higher in dividing cells since they are synthesising high quantities of DNA, and also of histones to pack it with. However, as seen in Figure 6E not all of them are enriched to a similar extent. The expression of chromatin regulators in stem cells has been associated with pluripotency by multiple studies. Here, we just wanted to report the enrichment that we observe, while we give specific enrichment values and p-values for each gene. Future studies will shed further light on the function of chromatin regulation in pluripotent stem cells.

- Vigilin⁺ cells, salivary glands and polyploidy. Nuclear size can also correlate with transcription activity without that implying polyploidy (e.g. <https://pubmed.ncbi.nlm.nih.gov/7523596/>), and as the authors acknowledge, these cells have high RNA counts per cell. Is it possible to experimentally validate the presence of multiple chromosomes (e.g. using colchicine) in these cells? That will solidify their interpretation. Regarding the evolutionary scenario (lines 356-358), salivary glands with polyploid cells appear largely restricted to some insect groups. Therefore, if demonstrated, polyploidy in this cell type would likely be convergent between annelids and insects and not a case of deep homology.

We thank the reviewer for such an insightful comment. We have added this possibility to our text, and cited the reference. We believe that the experiments that the reviewer suggests should be possible, but we would need more than 3 months to perform them and believe that they are beyond the scope of the current manuscript. We just wanted to present the data and hypothesise on their possible function. Regarding the evolutionary scenario, we believe that the hypothesis of deep conservation hinges on the shared expression of vigilin. We have clarified this point in the text, which also contemplates the alternative scenario by convergent evolution.

- While the dataset and findings are solid and high quality and largely justify the conclusions, the text lacks context and a broader picture (especially in the Intro and Discussion) and is sometimes repetitive. For example, while the abstract reads more like a series of findings, the last paragraph of the Introduction is more engaging and better addresses the overarching, and perhaps more generally appealing, question of the manuscript. Likewise, the Introduction focuses on the technical value of scRNA-seq, but it could provide a better picture of the knowledge gaps in stem cell biology and regeneration in annelids (what is known and needs to be solved). Some of the text in the Results section is redundant or could fit better in Methods, especially in the first section (e.g., sequencing mode, etc), and I wonder if Supplementary Note 1 could be incorporated within the main text, blended with some of the text from lines 152 to 236 under a broader section on cell type diversity. Indeed, it is unclear why the validation of epidermal, muscular and neuronal cell identities is a separate section when all main clusters are generally experimentally validated. Some aspects of the Results are also poorly explained. For example, in lines 262-264, when referring to the centrality of TFs in co-expression networks. What does that mean? What is the biological insight that derives from this analysis? Finally, the Discussion acts more as a Conclusion section, summarising and repeating the main findings rather than putting them in a broader context. For example, the end of the sentence in line 233 gives more or less the same idea as the following sentence (and see point above regarding polyploidisation) and lines 361 and 362 of the Discussion.

We have introduced several changes in the text to address these concerns:

- We have modified the abstract to shorten it.
- Most of the technical details about the analysis have been removed from the first section and included in a new Supplementary Note 1.
- A new paragraph in the introduction provides a better picture of the knowledge gaps in stem cell biology and regeneration in annelids.
- A new discussion figure (Figure 8) puts forward several hypotheses derived from the work and discusses future directions for this research.
- These additions, together with the new Figure 7 necessitate explanation in the main text, which is as a result somewhat long. We believe that bringing Supplementary Note 1 to the main text will contribute to making it longer
- We merged the epidermal, muscular and neuronal sections with the gut section

Regarding the TF centrality question, we provide here a more extended explanation. To convey the biological relevance of TF centrality, we have expanded our exploration of TF centrality in relation to another important WGCNA metric called module connectivity. Based on (Horvath & Langfelder, 2008), the metric of “connectivity to module” or $kME(x,y)$ is defined as the correlation of the gene expression of a given gene ‘x’ with the weighted average expression of all genes in a given module ‘y’. We reasoned this definition could be extended in the case of TF genes as the likelihood/potential of a given TF gene to play a role in the regulation of the expression of the genes in a module. This definition goes along the same lines of current standard methods to infer the regulatory capacity of a TF based on gene expression (ref ANANSE, ref SCENIC, ref. GRN Scalambria et al., ref. GRN Badia-i-Pompel et al 2023).

We reasoned that, if a graph is constructed where genes are connected based on WGCNA coexpression values (such as in Fig. 4E,F,G), a TF ‘x’ connected to many genes of a given module ‘y’ should exhibit a high centrality $C(x,y)$ as well as a high connectivity $kME(x,y)$ value. Therefore if there is such an agreement between the two metrics, we argue that centrality in a graph might also prove useful to identify potential regulatory TF candidates.

Since TF centrality is calculated independent of the kME , we explored the relationship between these two metrics. We observed that TF genes that survive the Connected Component analysis (i.e. those present in the graph of Figure 4E) and thus have a centrality value calculated, show an overall higher connectivity, as shown in the plot below:

This correlation between centrality and connectivity is intrinsic to each gene module, as every gene module behaves somewhat differently in terms of gene expression. Thus, high correlation values between kME and centrality only emerge when inspecting the relationship between these metrics for each module individually, as shown in the plot below:

Per-module correlation between TF connectivity and centrality

Provided most of the gene modules we found exhibit a strong, cell type-specific gene expression, we argue that it is possible to find potential regulatory TFs of cell type identity and function using both connectivity and centrality. We have generated a new supplementary figure to show these analyses.

Other points:

- Line 35: "cell type diversity of adult annelids for the first time"; by scRNA-seq approaches because many other methods have been used to characterise this animal group's anatomical and cellular diversity.

Changed

- Lines 94-96: how many of the transcripts have an annotation?

We have included this information in the main text.

"Of the 29,807 transcripts, we annotated 18,551 transcripts using eggNOG (Cantalapiedra et al., 2021) and 19,582 transcripts using Diamond Blast (Buchfink et al., 2021) (18,114 transcripts overlap, Supplementary File 1, Supplementary Note 1)"

- The authors use the terms counts and UMI interchangeably, but it might be better to be consistent.

We have unified the terms in "UMI counts" where possible

- Line 132: Figure 2E, missing closing bracket.

Changed

- Line 152: HCR in situ hybridisation should be in situ HCR.

Changed

- Line 165: the longitudinal fibres are not obvious from the HCR pattern; a phalloidin staining on a separate sample may help.

We have strengthened the visualisation by adding arrowheads. Unfortunately, phalloidin stainings are challenging with alcohol-based protocols such as the current *in situ* HCR protocol.

- Line 191: glial-like cells have been described in annelids (e.g., <https://royalsocietypublishing.org/doi/10.1098/rspb.2017.0743>), so their existence as a separate cell type is not new.

We thank the reviewer for pointing out this paper. We have checked the most important RNA marker described in that paper, and identified by BLAST the closest match in the Pristina transcriptome, PrileiEVm001886t1. Unfortunately, its expression in our dataset is not very high (see below), so it does not allow us to see if the glial cells reported by Helm et al are the same ones that we propose as a glial-like cluster by their morphology. However, we have cited the manuscript in that passage.

- Line 197: "We also found annelid types" should be "We also found clusters"?

Changed

- Line 255: eleocyte are mentioned here for the first time without proper description.

We have now included an *in situ* HCR experiment for eleocytes in Figure 3D, described them briefly in the related section and given appropriate references.

- Line 340: "species capable of extensive adult cell type generation and regeneration" is an odd expression. I guess the authors mean *Pristina* has a diverse repertoire of cell types that can be regenerated?

This sentence refers to the capacity of generating adult cell types both as part of the normal asexual growth by paratomic fission as well as by regeneration after injury. We have added a sentence to clarify this.

- Line 334: "this shows that epigenetic regulation is a conserved feature of animal pluripotent cells." Is this surprising? Epigenetic regulation occurs in all cells and is

The comment from the reviewer seems to have been cut. We find surprising the high expression of certain epigenetic regulators, which are expressed at higher levels than in any other cell type. We believe that this has functional consequences on the regulation of pluripotency. In order to respond to this comment we have: 1) slightly edited this sentence to state that what is conserved is "high expression of epigenetic regulators" and 2) further discussed this in the discussion section. We believe that our data will foster studies on the role of these regulators in pluripotent cells in different invertebrates.

- In figures and panels of HCR, wouldn't be better to refer to the gene being used for HCR with the standard orthology instead of the contig name? If there is an annotation, of course. It makes it easier and more informative for the reader.

Some of the markers do not have annotations, sometimes because they are not annotated and sometimes because they do not have identified homologues in other organisms. Furthermore, the transcript IDs are univocal and this will therefore facilitate the use of the sequences for further research. We give annotations for these markers in Supplementary Files 1-4.

Reviewer 2

Reviewer #2 (Remarks to the Author):

Alvarez-Campos et al. sought to describe the cell type composition and diversity in the annelid *Pristina leidyi* (referred to as *Pristina* further on) by means of computational analysis of a single-cell transcriptional atlas. Taking this approach, they characterized and validated the major cell types in *Pristina*, which includes gut, muscle, epidermal and neuronal cells. In addition, they also identified and validated annelid-specific cell types, some of which have not been previously reported (*Idlrr+* cells and carbohydrate metabolic cells). With the power of this cell type atlas, they also sought to investigate the putative stem cells in *Pristina*. *Piwi+* cells have been studied in *Pristina* in the context of the germ line, but given the role of *piwi+* stem cells in highly regenerative animals, they sought to investigate whether the *Pristina piwi+* cells share characteristics with potentially homologous cell types in other systems. Altogether, they present a characterization of the cell type diversity in *Pristina* and provide molecular data that suggest a stem cell-like role to the population of *piwi+* cells in *Pristina*.

This study is valuable as a resource as it provides a molecular characterization of the cell types in *Pristina*. Cell types atlases at this resolution have proven to be useful in many other invertebrates (planarians, sponges, cnidarians, acoels, etc), given its power to both validate their cell type diversity and open the ground for new hypotheses to be tested functionally. While the characterization of the major cell types in *Pristina* was validated through HCR in situ hybridization and confirms their hypothesized expression patterns, the study lacks functional characterization of the putative stem cell population in *Pristina*. Stem cells in the previously-studied invertebrate systems have been characterized by means of molecular and functional analysis— the statements about the distribution, potency and differentiation dynamics of *Pristina piwi+* cells remains to be validated using functional tools (HCR, IHC, EdU, etc). The single-cell RNA-seq data present an exciting hypothesis of these cells being stem cells akin to neoblasts or i-cells; however, molecular trajectories are not evidence of this property, they are hypotheses that need to be validated. This work is high-quality and will have high impact on the field, but it is very important for the authors to restate their claims to make clear that many of them relating to the *piwi+* population are hypotheses. It would also be important to clearly outline alternative hypotheses that could explain the data. This does not weaken the paper but instead strengthens it; it lays out a roadmap for important work to be done next. See below for details on these major comments.

We thank the reviewer for their insightful comments. We have extensively revised our statements throughout the manuscript and as detailed below. Chief among these revisions is a new discussion figure (Figure 8), where we outline the distinct hypotheses that emerge from our data and discuss them with the available literature.

1) Issues with the UMAP and PAGA plots in Figure 1

The current clustering of cells in Figure 1B, while being clear in sense of the chosen color-code, lacks organization in the annotation of some clusters. In some instances, it is complicated to determine 1) which clusters are annotated and which ones are not, and 2) what cluster numbers belong to which clusters. Here are two examples:

- Overlap between cluster numbers [such as 47 overlapping a number below it] clouds the assignment of these cluster numbers to their respective clusters.
 - The presence of more than one cluster number too close to each other [such as 45 and 2 on the same gray cluster], making it hard to identify what clusters those numbers belong to.
- This becomes an issue when trying to understand the PAGA graph in Figure 1D. Given that the PAGA algorithm reconstructs lineages by means of the topology of a given UMAP, it is hard to visually assess the vicinity of a cluster to its connected clusters. For instance, cluster 2 connects with clusters 21, 34 and 41, but it is complicated to see where those clusters are in Figure 1B. Overall, organizing labels and annotating the clusters properly will allow for a streamlined understanding and takeaway of the UMAP.

In order to improve the visualisation of clusters in Figure 1B we have: a) removed the “unannotated” cluster labels (46, 47, 48, 50, 51, 52, 53, 54, 56, 57, 58). These clusters are mentioned in the text and can be seen in Supplementary Figure 3. b) manually separated overlapping labels such as 45 and 2. Supplementary Figure 3A expands the annotation of individual clusters within each broad group. We have tried inserting all labels in the UMAP visualisation of Figure 1B but then these labels overlap too much and it is more difficult to identify each cluster.

2) Functional validation of piwi+ cells in Pristina is lacking

While Pristina piwi+ cells have been explored in the context of gonadal dynamics in Ozpolat & Bely 2015 and in Ozpolat et al 2016, the present study lacks a clear definition and validation of the Pristina piwi+ cells in terms of their putative identity as stem cells. Figure 5A suggests that these piwi+ cells are proliferative, based on the expression of proliferation markers (pcna, h2b, h2a) in these piwi+ cells; validation of this could be provided by developing a double piwi HCR in situ with the labeling of the mitosis marker H3P. This experiment would not only confirm these cells to be proliferative, but also provide information about the proportion of piwi+ cells that are proliferative and the distribution of these cells along the Pristina body plan. Further, if these cells are proliferative, the EdU protocol published in Ozpolat et al 2016 could be used to trace EdU+ cells (which would presumably also be piwi+ cells) and ask whether they differentiate into multiple cell types. These experiments would allow to explore the cellular properties and differentiation potency of the Pristina piwi+ cells. The goal of mentioning these experiments is not to suggest that the authors must do them all for the context of this manuscript, but to underscore that claims about the properties of this cell type, if it is one cell type at all, need to be supported with experimental evidence of its existence, its distribution, and its proliferation and differentiation properties. Without that, all statements need to be tempered and presented as hypotheses. Key things that need to be added/modified:

- An explanation of whether HCR was attempted on piwi and all the other markers of the piwi+ population found in this study.
- Given that there is no corroboration provided on which cells express these markers and where they are located in the Pristina body plan, it is important to consider that Pristina might be more similar to vertebrates in that there may be distinct tissue-restricted specialized stem cell populations that are unified in their expression of proliferation genes, etc. This is a very different scenario from them being pluripotent stem cells like neoblasts or i-cells. This scenario would look indistinguishable from the pluripotent population scenario in terms of single-cell RNA-seq. Therefore, the authors should include this and any other alternatives as they interpret their data.

We have introduced several changes in the text to address these concerns:

- We have now performed EdU labelling experiments to identify proliferative cells and colocalised these cells by *in situ* HCR with a marker of *piwi*+ cells, the histone 3 (h3) homologue PrileiEVm022498t1. This experiment confirms these cells to be proliferative, and also provides information about the proportion of *piwi*+ cells that are proliferative and the distribution of these cells along the *Pristina* body plan, as the reviewer requested. Further work using these markers will elucidate the cellular and molecular dynamics of these cells in growing conditions as well as in regeneration.
- We have clarified in the text that we have attempted to localise *piwi* by *in situ* HCR, and that we chose the *h3* marker as a stronger probe that is easier to interpret.
- We have discussed the different possible scenarios in a new discussion figure (Figure 8) and the accompanying text.

3) Overinterpretation of PAGA results

The authors rely heavily on their graph abstraction algorithm to make conclusions about the putative stem cell population. PAGA is very useful for quantifying interconnectedness of clusters, but it is important to remember that ultimately that interconnectedness is only based on gene expression similarity. Gene expression similarity can be suggestive of a differentiation trajectory, but it cannot serve as evidence of differentiation. It is interesting to note that several small clusters were recovered close to the putative stem cell clusters, and therefore it is no surprise that PAGA then shows the stem cell cluster as being connected to more other clusters than any other cluster. This is not at all convincing as evidence of potency. See e.g. Lines 289-290: “The highest potency score in our abstracted graph was attained by *piwi*+ cell cluster (Figure 5D), strongly suggesting that *piwi*+ cells are pluripotent stem cells.”

We apologise for the confusion. We never claimed that the potency score gives evidence of developmental potency, and we only used it as a model. We have now better explained this in the text including the sentence “*While showing the developmental potency of a cell population necessitates transplantation experiments, the potency score is a useful model to hypothesise it from single cell expression data*”. We have toned down the following statement: “*The highest potency score in our abstracted graph was attained by *piwi*+ cell cluster 2 (Figure 5D), suggesting that *piwi*+ cells may be pluripotent stem cells*”. In planarians (Plass *et al.* Science 2018), the potency score was highest in the neoblast population, which had been previously shown to contain pluripotent cells by transplantation experiments. We have now performed this same modelling exercise in *Pristina* cell populations and the result is that the *piwi*+ cells attain the highest score. We believe that this is worth reporting.

4) Statements and inferences are not supported by the data presented

Along the results and discussion sections, particularly in relation to the *Pristina* *piwi*+ cells, the authors’ conclusions and inferences based on the data presented are often not consistent.

Providing data supporting the hypothesis that one cell population is pluripotent, and thus able to give rise to all cell types in *Pristina*, requires in vivo lineage tracing, which was not presented in this study. Therefore, statements regarding the pluripotent identity of cells in *Pristina* should be edited to convey what the data really presents. Below are some sentences that should be edited (this list is not exhaustive and authors should revise other related statements):

- Lines 281-283 “Moreover, our PAGA analysis revealed that most differentiated cell types were connected to piwi+ cells by differentiation trajectories (Figure 1D, Figure 5B), including epidermis, muscle and gut, suggesting these cells are a pluripotent population.”. Once again, lines drawn by PAGA are really only showing quantified interconnectedness in terms of gene expression similarity which can serve as hypotheses of differentiation. The lines should not be interpreted as de facto differentiation trajectories. Yes, PAGA incorporates RNA velocity, but there is simply no way to know in an unvalidated dataset how effectively that is capturing differentiation.

We have further stressed that our use of “differentiation trajectories” is to indicate computationally reconstructed differentiation trajectories using a lineage reconstruction tool. Now all instances where we use “differentiation trajectories” contain the word “reconstructed” or “reconstruction” to clarify. This is in line with other papers in the field.

- Lines 289-290: “The highest potency score in our abstracted graph was attained by piwi+ cell cluster (Figure 5D), strongly suggesting that piwi+ cells are pluripotent stem cells.”

This has been rephrased as explained above

- Lines 303-304: “Altogether, these analyses showed that piwi+ cells in *Pristina* are a heterogeneous cell population with a pluripotent stem cell signature.”

The statement does not say that piwi+ cells are pluripotent, but that they have “a pluripotent stem cell signature”. We clarify in the text what we understand by “pluripotent stem cell signature”: a series of observations found in *bona fide* pluripotent stem cells such as planarian neoblasts. We have clarified this in the text.

Lines 334-337: “Taken together, our data strongly suggest a model where post-transcriptional and epigenetic regulators control stem cell maintenance and pluripotency, and a panoply of TFs prime these to differentiate into multiple cell types.”

We have toned down this statement, but note that this is also phrased as a suggestion.

Minor comments

- Lines 56-58: Provide reference for other single-cell atlases: from planarians (Fincher et al 2018) and acoels (Duruz et al 2021, Hulett et al 2023)

Added

- Lines 75-77: Sentence feels incomplete

We have slightly edited this sentence and provided references

- Lines 142-143: It would be helpful to specify how the PAGA algorithm works and why in this case is more helpful to use PAGA over other algorithms to reconstruct the differentiation trajectories.

We have added an explanatory sentence about PAGA. We do not intend to say that PAGA is better than other algorithms and extensively benchmark it is use.

- Line 265: Centrality of a TF could also suggest their importance in the functionality of the mature cell types, not just only their differentiation.

Changed

- Line 300-301: Specify what subcluster this is

We have added one explanatory sentence about subclusters

- Figure 1D and Supplemental Figure 4: It is unclear why there are some clusters that appear in Supplemental Figure 4 but not in Figure 1D (for example, cluster 48, 57 and 58 are in Supp4 but not in 1D)

We performed this analysis without the “unannotated” clusters (46, 47, 48, 50, 51, 52, 53, 54, 56, 57, 58). The rationale behind this is explained in Supplementary Note 1.

- Figures 3B, 3C, 3H, 3G, 3I: Are the extensions in the outside of the body plan also positive for their assay, or is the staining considered background?

These are the chaetae, which are visible very brightly in most channels. This is a case of autofluorescence, not a real signal. We explain background signal and autofluorescence more in detail in Supplementary Figure 6, by comparing real samples with negative controls. For further clarification, we have added this sentence to the supplementary legend: The chaetae (green, hollow arrowhead) is also shown as an example of intense autofluorescence.

- Figure 5C: Specify the names of the transcripts in the dot plot

Changed

- Supplement Figure 9: Specify the transcript names of each of the UMAP projections

These are not individual transcript UMAP projections, but scored sets transcripts. The score represents the aggregated and averaged gene expression of all markers collectively. The scoring function is described in the methods section “Scores”.

Reviewer 3

Reviewer #3 (Remarks to the Author):

The work under consideration represents a convincing effort to understand the cellular landscape in an asexually-reproducing and regenerating clitellate annelid, the oligochaete *Pristina leidyi*. A total of 75k cells have been transcriptome-sequenced with the SPLiT-seq method. The authors are to be commended for producing a technically sound and interesting dataset that will serve the *Pristina* and annelid community to advance on the expression characterization of stem cells and differentiated cell types in this and other annelids. In its current state, however, the analysis remains rather shallow and does not go beyond a cursory identification of markers for a subset of the identified cell clusters, mostly at a rather low resolution such as 'epidermis', 'muscles', or 'neurons'. For the gut, the authors go into more detail and reveal a striking anterior-posterior regionalization of the gut into different cell populations with sharp boundaries that are nicely documented by HCR in situ hybridization. Most emphasis is on the presumed stem cell fraction of the single cell dataset, which is analyzed via partition-based graph abstraction (PAGA) of the connectivity between cells. This identifies a population of cells connecting to many possible differentiation trajectories and thus most likely representing pluripotent stem cells. This is corroborated by the expression of proliferation and chromatin regulators in these cells. Overall, this is an important addition to the growing number of single-cell datasets in different animal phyla, and will represent an important comparative resource for the regeneration and evolutionary cell biology communities. While it falls short of the in-depth analysis and identification of cell types, it opens up a new level of understanding of pluripotent progenitor and stem cell populations in an important new model for asexual reproduction and regeneration, complementing the study of planarians.

We thank the reviewer for their praising comments and their insightful suggestions below. They have helped shaping our description of the WGCNA analysis among other as detailed in our point by point response.

Some points for improvement in order of decreasing relevance.

- It is not clear why there is not much of a comparison to the Schmidtea data and how the different datasets compare; what is the relationship to Schmidtea neoblasts, their potency and differentiation potential, their transcription factor and regulator signature etc. These data are available and a comparison is straightforward using SAMap, data integration integration approaches or alike. Schmidtea shares with *Pristina* the major characteristics relevant here, i.e. asexual reproduction and high regenerative potential.

Following the suggestion of the reviewer, we have compared *Pristina leidy* clusters to *Schmidtea mediterranea* clusters using SAMap. We present the results of the analysis below. This was an interesting and reassuring comparison as indeed our *Pristina piwi+* cell clusters show the highest resemblance with planarian neoblast populations. We also retrieved high scores between cell types of similar annotation (i.e. epidermal cells with epidermal cells, muscle cells with muscle cells, etc.). This can be visualised in the heatmap of SAMap scores between pairs of clusters.

At the same time, when we explored the SAMap score values of *piwi+* cells of *Pristina* with all the *S. mediterranea* cell types, the highest similarity scores were with *piwi+* cells of *S. mediterranea*. This can be seen in the box plot below, where we see SAMap scores of annelid *piwi+* cells vs planarian *piwi+* cells are much higher than SAMap scores of annelid *piwi+* cells vs planarian non-*piwi+* cells.

However, there are several reasons for excluding this analysis from the current manuscript. First, we consider this is beyond the scope of the paper, in which we aimed to present the different cell types of an adult annelid rather than performing inter-species cell type comparisons. Second, there are some other interesting aspects in this comparative analysis that are worth exploring more in detail in order to discern true homology from similar functionality, such as the case of ciliated cell types. These cells tend to share high SAMap scores between species, which is correlated with a similarity on gene expression rather than to the true homology between clusters. Therefore, we consider this sort of analysis may be more suitable for a future piece of work.

- The study is technically high-quality, including for example a new, rather complete long-read reference transcriptome, stringent filtering, and high cell numbers that help overcome the low numbers of UMIs/cell. It is less clear what exactly has motivated the clustering cut-off but but the clusters defined by the authors appear sufficiently differentiated for this to not be critical. However, the homogeneity of the clusters presented in Figure 1B would be more accessible if instead of feature plots as in Fig. 1C, a heat map would be shown for the 50 most specific genes for each cluster and their expression across all cell types (an extended version of the dotplot in Suppl Fig. 1E). Since the authors were more interested in (re)generation, the in-depth annotation of the cell atlas is of lesser importance.

- For the transcription factor analysis in Fig. 4, the authors' approach to identifying TFs is valuable and much appreciated. It is a pity though that the identity of the TFs defining the

clusters cannot be accessed. Fig. 4C should be presented as a “staircase” plot as in 4A and TF names be added (e.g., in a Suppl. Fig.). The TFs and the table underlying Fig. 4C should be made available. As above, the feature plots presented in Fig. 4D are the least favorable solution to illustrate TFs versus clusters and could go to supplementary material, replaced by dotplots. How many cells express the TF per cluster?

We thank the reviewer for these suggestions. We have rearranged the rows of the heatmap in Fig. 4C to highlight the TFs most highly expressed on each cell type, effectively creating a “staircase” plot. We have updated Supplementary Figure 9 to include, alongside the original columns of TF evidence: (i) a yes/no column of whether a TF is present in figure 4C, since not all predicted TFs are well expressed in our dataset; (ii) a column that tells in which cell type this TF has the highest expression; and (iii) the expression values (in counts per million, cpm, see Methods) for the TFs present in Figure 4C.

Regarding the dotplots, plotting the expression of single cell data is challenging. We have produced a dotplot for the TFs in Figure 4D, as the reviewer suggested. We find that they convey the point of the figure in a way that is less satisfactory than the UMAP plots provided. This dotplot can be seen below:

In essence, the ratio of expression in *piwi*+ cells vs the rest of the cells varies for each of the TFs. This creates problems in the dotplot visualisation. So, for instance, the first one ('PrileiEvm000093t1') has a comparable ratio of expression in *piwi*+ cells and the differentiated type highlighted, epidermis. This results in circles that are largest in these 2 broad types. However, for the second ('PrileiEvm003917t1'), the expression in the neurons achieves a higher ratio of the expression, and dictates the size of the largest dot. The second largest are polycystin cells, which the UMAP also correctly displays, but the size of the *piwi*+ cells is very small, compressed by the relatively high expression in neurons. That is the case for *myoD*, the muscle TF ('PrileiEvm008071t1'). A third layer of confusion is happening due to the different sizes of the broad groups, and the shallowness of the data. For instance, for the gut factor *hnf4* ('PrileiEvm006891t1') the UMAP shows expression in the *piwi*+ central area as well as several of the gut types. In the dotplot we also see a large dot in the carb. metabolic cells, arginase+ cells and *ldlrr*+ cells. This expression is true, but these dots achieve the same size in the dotplot because what is plotted is the fraction, but there are a lot less cells in these 3 clusters than in the *piwi*+ and gut clusters. We think that the UMAP plots convey the message, i.e. that there is expression in *piwi*+ and gut in a more intuitive way. Indeed, part of the problem is that these TFs are expressed very lowly, but this is precisely why we resorted to a pseudobulk assay in Figures 4A-C and Figure 4D serves as a validation of this to show the data in a raw format.

- The graph analysis is confusing. WGCNA by design groups genes that show correlated expression; it should be no surprise that clustering a graph made from the TOM matrices, which are needed to cluster the genes into modules in WGCNA, shows the same thing (“[...] identified several connected components that reliably match the WGCNA modules”). Furthermore, not having seen this type of analysis elsewhere, is there a more straightforward way to arrive at a similar conclusion, e.g. by just calculating correlation between all WGCNA genes and placing cutoffs on that to form a graph?

We thank the reviewer for this comment. Indeed our work is one of the first WGCNA analyses for whole organism cell atlas exploration and we believe that our analyses will be useful for other researchers. We explored a potential graph generated from the Pearson Correlation Coefficient (PCC) values between pairs of genes, and selecting for PCC values above 0.675 as seen by the histogram below:

We plotted this network using the Kamada-Kawai layout algorithm, the same one used in Fig. 4E, and retrieved the following layout:

As we can see there is a certain level of information present in the method proposed by the Reviewer. We attempted to retrieve connected components as done in the analyses of our manuscript, but failed to retrieve more than one component with a sizable number of genes. This is shown in the plot below, where we see that the first and largest connected component contains the majority of the genes in the PCC graph. We argue that this is due to the high interconnectivity derived from the metric used to generate this graph (PCC values):

Because of the high graph interconnectivity derived from PCC, we think more developed methods are required to discern the structure of the data (i.e. how genes are connected based on their level of gene expression) and this is where we think WGCNA provides a viable approach (with its Beta parameter to discern spurious correlations, and with the Topological Overlap algorithm that reinforces coexpression-derived connections based on the number of shared neighbours). This is absent from a single-step approach like PCC.

Regarding the initial question, we apologise for the confusion and have attempted to better explain the meaning of each plot in our revision. We would like to raise the point that, despite the same starting material (the TOM matrix), the methods underlying WGCNA's algorithm for module membership assignment (using the `cuttreeDynamic` function from WGCNA) and igraph's connected component assignment (using the `CC` function from igraph) are independent (as in, they run independent to each other and they run following different procedures independent to each other), and they still reach a similar consensus (as shown by the adjusted Rand index between connected component membership and module membership, see below). Thus we see our graph analysis as an alternative approach that serves as validation of the results observed from WGCNA. We have provided an additional supplementary figure, new Supplementary Figure 9, where one can see the connected component composition in terms of the number of genes of each gene module (Supplementary Figure 9 A,B).

- The network visualization of the WGCNA modules shown in Fig. 4E appears to produce a measure of similarity between cell types by means of shared gene co-expression. How is this

going beyond what is already shown in Fig. 4A? The stomach-neurons-secretory-intestine-cluster in the 5 o'clock portion of 4E is surprising, as Fig. 4A does not seem to indicate much co-expression between at least most of these modules. What is the significance?

Again, we apologise for the confusion. We would like to raise the point that the main purpose of Figure 4E is to visualise the presence of different, sizable connected components in the network; i.e. the connections between genes from the same or related modules, and not the discerning of cross-module connections (or cross-connections as we call them in the manuscript). For the 4E plot we subsetting the gene connections in a somewhat restrictive manner (> 0.35) and we proceed with the network visualisation. To detect and explore module cross-connections in the 4F plot, we subsetting the graph using a lax threshold (> 0.2) and performed a different analysis. Thus, the significance of figure 4E is the exploration of the presence of connected components in the structure of the graph, a pre-requisite for Figure 4F as it would not make sense to try and seek connections between modules if these were not grouped in connected components and the whole network was fuzzy.

The merge of different modules at the five o'clock portion of the layout is due to an artefact of the Kamada-Kawai (KK) layout algorithm. Representing graphs containing tens of thousands of nodes in 2D is challenging. The KK algorithm uses a spring-like method to disperse graph nodes (i.e. genes) in a confined space. This can lead to visual artefacts where different nodes appear to be interwoven with each other when they are in truth not connected. When subsetting the Figure 4E network to just contain the modules mentioned by the Reviewer, we can see that there are indeed no cross-connections between neuronal and intestine modules and that these are in fact separate connected components. This can also be seen when looking at the module composition of each connected of these components (of note, one single gene of one secretory module belongs to another connected component and this is why there are two purple bars in the plot on the right):

five o clock portion of the graph

Genes of each module in the 5 o clock belong to different connected components

We have replaced the previous plot of 4E for a new one using the Fruchterman-Reingold layout algorithm, which we believe produces a layout more representative of the graph. You can find it as a new Fig. 4E and also here below:

Pristina network, color of cell type at which max expr, Fruchterman-Reingold Layout

- The authors mention that they used arbitrary cut-offs to build the WGCNA graph. While many score distributions follow power laws, sometimes the underlying distribution is bimodal. Did the authors investigate that?

We would like to thank the Reviewer for this comment which has proven insightful to enhance and expand our knowledge. Firstly, we apologise for the use of the word “arbitrary”, when we ment “empirical”. We first explored the behaviour of the network by looking at the distribution of connection values. We realised there was an exceedingly steep slope of very low values in the distribution, which correspond to background connections of all vs. all. To further inspect the network after removing these interactions and keeping the tail values of the distribution, we defined a set of increasing threshold values to try and iterate the generation and visualisation of a graph. This code and the respective graph layouts can be found in our GitHub repository. Upon visual inspection of these graphs, we settled for using 0.35 as the threshold value for Fig. 4E and the Connected Component analyses.

Thanks to the reviewer’s suggestion, we further inspected the behaviour of this score distribution and, despite an initial power law-like behaviour, we were able to detect a smaller second mode right at the interval between 0.35 and 0.4, which is reassuring of our initial decision to employ a threshold of 0.35. This can be seen in the figure below, showing the distribution of one million of TOM values at four different cutoffs: 0, 0.01, 0.1, and 0.2. The red line in the last histogram correspond to 0.35:

In addition to all of these responses, we have included a Supplementary Note 3 file with a more detailed description of these analyses.

- The claim that vigilin+ cells are polyploid cells that function as glands is very speculative and should be toned down.

- The PIWI analysis is in the heart of the study. Given that, it will benefit from some additional detail. Did the authors try to expand the PAGA analysis to the subclustered piwi+ cells (all 8 subclusters from the supplementary figure)? This might offer some clarity that the UMAP cannot provide, also with regard to the possible pluripotency of cells. Also, the identity of the TFs expressed in Piwi+ cells should be revealed and how they relate to the subclusters.

We have tried to perform PAGA in the subclustering experiment. However, we think that we are reaching the resolution of this analysis. The interpretation of the PAGA graph (below) heavily hinges on its rooting point. Here we chose “0” arbitrarily, but there could be less committed. Further work is needed to determine this.

We also include here the TFs plotted in Figure 6D. This analysis is interesting, and consistent with the interpretation. Indeed, these TFs are heterogeneously distributed throughout the subclusters. However, some of these TFs are very lowly expressed in the *piwi*⁺ cell population.

Consistent with the interpretation made in the text, TFs that are expressed in differentiated gut (PrileiEVm008001t1, PrileiEVm010521t1, PrileiEVm007974t1) are expressed in subcluster 3 and 1, whose markers also score high for gut expression. Similarly, PrileiEVm008837t1, expressed in nidogen+ cells and muscle, is high in subcluster 4, whose markers score high for the nidogen+ cells. Indeed, our PAGA analysis in Figure 1D connects the nidogen+ cells with muscle clusters in a differentiation trajectory. We believe that this sort of analysis will be fundamental in future annelid stem cell research, but however it is very preliminary at this stage.

With this analysis our main point is about the heterogeneity of piwi+ cells, which is a solid finding that arises from several sources of evidence. Future work will resolve further the piwi+ cell subclusters, at the transcriptomic and at the spatial level

- Polycystin not “cistin”

Changed

REVIEWER COMMENTS

Reviewer #1 (Remarks to the Author):

This revised manuscript, "Annelid adult cell type diversity and their pluripotent cellular origins", is an improved version and an enjoyable read. The new data and analyses clarify most of the concerns I raised before. However, there are some points the authors might want to consider:

- The authors toned down their interpretation of vigilin+ cells as potential salivary glands. However, I do not think they provide (new) evidence to keep suggesting these might be polyploid and a case of deep homology with those of insects (i.e., that the last common protostome ancestor had a salivary gland cell type expressing vigilin and mucins). Vigilin is involved in many cellular processes (as REF 56 demonstrates), and many possible reasons can cause larger nuclei. Indeed, the reference they use to argue for the role of vigilin in controlling ploidy in *Drosophila* (REF 57) demonstrates an association with heterochromatin, not with polyploidy in this insect (the role in ploidy is in yeast). The authors' interpretation of this cell type is widely speculative. Their data detect a distinct cell type that morphologically, cytologically, and transcriptionally could be a (salivary) gland. This is an interesting finding, and further work will confirm what this gland is and, if it is a salivary gland, the potential origin and diversification of these in Annelida and animals in general. I suggest rewriting lines 166-179 and 343-348.

- The author's efforts to colocalise piwi+ cells with lineage-specific TFs are commendable. However, it is unclear why the authors chose those TFs. What is their orthology? Are they likely to be upstream in the lineage-commitment cascade? A bit of context would help strengthen their claim that piwi+ cells expressing certain TFs are the initial progenitors of the different cell types. Likewise, the authors decided to maintain the gene model name in most of their figures, but that really affects the interpretation of the data (perhaps adding both the gene model name AND orthology in the figures helps?). PrileiEvm025662t1 (neuronal) is also very broadly expressed in the fission zone. Is that normal for a neuronal marker? Are some of the PrileiEvm025662t1 actual neurons?

- Is it possible that there are lineage-committed stem cell populations that are piwi- (e.g. epidermal) that do not come from a piwi+ stem cell pool? The authors kind of suggest it in the discussion, but it is not really in Figure 8.

- Line 85: "we characterised the molecular signature of annelid piwi+ cells at both the transcriptional and epigenetic level". The authors do not technically characterise piwi+ cells at the epigenetic level. They assess the expression of epigenetic regulators in piwi+ cells, but we do not know how these cells differ epigenetically from the rest of differentiated cell types. I suggest rewording as "we characterised the molecular signature of annelid piwi+ cells at the transcriptional level"

- Line 160: "involved" instead of "including"?

- Lines 232-233: "and showcases the power of ACME and SPLiT-seq to unravel the TF expression underlying cell type differentiation or function in uncharacterised species." This is an odd sentence, especially when the authors admit in the first sentence of that paragraph that their combinatorial single-cell dataset had a low UMI and gene count, and thus, they decided to use a pseudobulk approach to investigate transcription factor expression. I suggest deleting the phrase in lines 232-233.

- Line 260: change "reinforcing the notion" to "reinforcing the scenario"

- Line 372: "piwi" in italics at the end of the line.

Reviewer #1 (Remarks on code availability):

The repository seems to be complete and well explained

Reviewer #2 (Remarks to the Author):

Authors have revised and implemented all of the points that were raised carefully— they have expanded their hypothesis framework, spearheaded by the addition of the new discussion figure (Figure 8), which highlights the diverse hypotheses that are emergent from their findings. In addition, authors performed EdU experiments that explore the cellular properties, including their proliferative identity, proportion and distribution in the *Pristina* body plan. Finally, authors understood the issues raised with language in the manuscript, mainly with the “reconstruction of the differentiation trajectories” using PAGA.

While these revisions certainly strengthen the the manuscript, it is important that authors detail all possible hypotheses in the text; the presence of the discussion figure helps with the conceptualization of the hypotheses that emerge from the study, but it is important to not favor any hypothesis over any other one in the text. For instance, the sentence “Altogether, these analyses showed that *piwi+* cells in *Pristina* are a heterogeneous cell population with a pluripotent stem cell signature that is also observed in other pluripotent stem cells.” is presented as one that only considers the hypothesis conceptualized in Figure 8A, but does not consider hypotheses 8B, 8C, or 8D.

Balancing the emergent hypotheses presented in this paper will allow for room to find commonalities in other annelid studies, such as the Stockinger et al manuscript that will be co-published with this article— stressing one hypothesis over any other ones will limit these commonalities. Although these papers focus on different species, and it is entirely possible that different cellular dynamics operate across annelids, given that the data in this manuscript cannot rule out multiple hypotheses, it is important to not leave the reader with just one major interpretation. For example, given that functional perturbation of the *piwi+* population has not been shown, it is a stretch to definitely state that the *piwi+* cells “underlie” differentiation, as stated in the abstract. The last sentence in the abstract could be reworded to read “Our data reveal the cell type diversity of adult annelids by single cell transcriptomics and suggest that a *piwi+* cell population with a pluripotent stem cell signature is associated with adult cell type differentiation.”

Reviewer #3 (Remarks to the Author):

In general, the authors have engaged with our suggestions, which is very positive. Some open questions/points remain however.

For the TF analysis in Fig.4, the authors did not address my concern, as the TFs best representing each cluster can still not be identified. Please add a plot with names at least to the Supplementary material.

Also, I am not sure about the insistence on the UMAPs: for example, the dotblots shown as text figures cannot convey the message because “gut” is only featured by one column. Do the TFs by any means relate to the AP sections of the gut? I insist that while the dot plots might overemphasize cell types with relatively few cells, at least we are getting some true data regarding specificity for a given type, for which it does not matter how frequent that type is, and that is the purpose of the figure. UMAPs are usually not chosen because colored dots are often plotted on top of each other and are hence invisible depending on the population density of that cluster within expression space - you are trying to compress 80k points in 5cm x 5cm pictures.

For *PrileiEvm000093t1*, a zinc finger, the UMAP suggests some expression in the *piwi+* cells and in the general epidermis part, with very few random occurrences of expression elsewhere; the dotplot however suggestssignificant expression e.g. in 10-20% of *lumbrokinase+* cells or *ldrr+* cells. In

general, the dotplots illustrate much better that the piwi+ cells share significant expression of multiple TFs with different clusters. I would thus urge usage of dot plots; alternatively, the authors may consider domino plots. –

The authors did quite some work on the former Fig. 4E, clarifying that the apparent connectivity between stomach-neurons-secretory-intestine was an artifact indeed. This is a significant improvement.

I would encourage the authors to include at least some of the extra work in some form, either as supplementary notes or as notebooks on their GitHub repository (which is finally public).

It remains unfortunate that the most interesting point, the heterogeneity of the piwi+ cells, does not emerge from the data more clearly, and that the double HCRs did not work so well.

REVIEWER 1

Reviewer #1 (Remarks to the Author):

This revised manuscript, "Annelid adult cell type diversity and their pluripotent cellular origins", is an improved version and an enjoyable read. The new data and analyses clarify most of the concerns I raised before. However, there are some points the authors might want to consider:

- The authors toned down their interpretation of vigilin+ cells as potential salivary glands. However, I do not think they provide (new) evidence to keep suggesting these might be polyploid and a case of deep homology with those of insects (i.e., that the last common protostome ancestor had a salivary gland cell type expressing vigilin and mucins). Vigilin is involved in many cellular processes (as REF 56 demonstrates), and many possible reasons can cause larger nuclei. Indeed, the reference they use to argue for the role of vigilin in controlling ploidy in *Drosophila* (REF 57) demonstrates an association with heterochromatin, not with polyploidy in this insect (the role in ploidy is in yeast). The authors' interpretation of this cell type is widely speculative. Their data detect a distinct cell type that morphologically, cytologically, and transcriptionally could be a (salivary) gland. This is an interesting finding, and further work will confirm what this gland is and, if it is a salivary gland, the potential origin and diversification of these in Annelida and animals in general. I suggest rewriting lines 166–179 and 343–348.

We appreciate the reviewer 2 comment noticing our error with the references mentioned in the text, that lead us to a misinterpretation of the results. We have already rewritten the paragraphs by the following ones:

lines 166–183: Then, we identified a cluster (23) marked by the expression of vigilin, an RNA-binding protein important for chromosome stability and cell ploidy⁵⁶. In *Drosophila* and humans, the vigilin homologue, DDP1, interacts with mRNAs localised in the endoplasmic reticulum^{57,58}. Pristina vigilin+ cells are located in three large bulbs in the anterior segments of the worm (Figure 3F). Based on their location and morphology, these likely correspond to pharyngeal glands, which have been described in many oligochaetes, including species of *Pristina*^{59,60}. Interestingly, this cluster showed a higher number of RNA UMI counts per cell (Supplementary Figure 2D). We wondered if this was a technical artefact or a biological observation instead, with vigilin+ cells being larger cells. We quantified the cell nuclei area of vigilin+ cells and determined that their size is significantly larger than that of other cells (Figure 3E). This large size could be a product of polyploidisation, but could also be a consequence of increased transcriptional activity or a higher amount of open chromatin⁶¹. Furthermore, we found a transcript encoding a mucin gene in the marker list. Together, our results characterise this cell type as pharyngeal glands from morphological, cytological and transcriptional data, but this interesting finding would require further work in order to suggest their potential function and diversification within Annelida.

lines 356-361: For instance, we found a *vigilin*+ cell type that expresses mucins and is localised in the head region, indicating that these are *Pristina* pharyngeal glands, previously described in other oligochaeta species. Interestingly, *vigilin* has been implicated in polyploidisation events⁵⁶ and we show that *vigilin*+ nuclei have larger sizes, consistent with a plausible polyploidisation. Nevertheless, further analyses would be necessary to confirm our hypothesis and to elucidate the function of this cluster.

- The author's efforts to colocalise *piwi*+ cells with lineage-specific TFs are commendable. However, it is unclear why the authors chose those TFs **(1)**. What is their orthology? Are they likely to be upstream in the lineage-commitment cascade? A bit of context would help strengthen their claim that *piwi*+ cells expressing certain TFs are the initial progenitors of the different cell types. Likewise, the authors decided to maintain the gene model name in most of their figures, but that really affects the interpretation of the data (perhaps adding both the gene model name AND orthology in the figures helps?) **(2)**. *PrileiEvm025662t1* (neuronal) is also very broadly expressed in the fission zone. Is that normal for a neuronal marker? Are some of the *PrileiEvm025662t1* actual neurons? **(3)**

(1) We think there is a bit of a misunderstanding here. The lineage-specific genes used in colocalization experiments with *piwi*+ cells (Figure 7) were not selected as TFs. Instead, they were selected as markers of distinct cell types:

'For this, we performed double in situ HCR using markers of piwi+ cells combined with top markers of differentiated cell types and EdU labelling of dividing cells' (line 314).

Marker genes typically include few TFs (because these have a rather low expression), but include the genes that define the clustering with high and specific expression. Usually, marker genes have an expression pattern that is restricted - or mostly restricted- to a certain cell type or cell group. Our goal using markers from differentiated lineages in combination with *piwi*+ markers was to prove that *piwi*+ cells express markers of cell types from different adult lineages:

'These results validate that piwi+ cells are a heterogeneous cell population, with a portion of the cells coexpressing markers of at least three different lineages, and a high proliferation rate in the adult stage. Taken together, these results suggest that piwi+ cells in Pristina are actively differentiating into diverse cell types in the adult worm.'

The selection of our marker genes went as follows: First, we selected a list of potential candidates for *in situ* HCR based on the visual inspection of the UMAPs of top markers (Supplementary Files 3-5). We chose genes that were specific to a cell group but were also detected in *piwi*+ cells (and this is the observation we sought to validate). After that, probe design constraints narrowed down the number of candidate genes. Finally, not all probes worked experimentally or showed clear signals. We select some of the best experimental examples for Figure 7: *PrileiEvm022781t1* (gut marker), *PrileiEvm025662t1* (neurons and polycystin) and *PrileiEvm008287t1* (epidermis). We also tried to colocalize *piwi*+

cells and muscle, but the *in situ* were not clear due to technical problems (weak signal, background noise, etc).

(2) Regarding the nomenclature and orthology, we kept the gene model names because markers PrileiEvm022781t1 and PrileiEvm025662t1 do not have diamond blast hits (Supplementary File 1). Marker PrileiEvm008287t1 shares orthology with multiple intermediate filament proteins. We have now specified this orthology in the main text and the figure legend. Our *piwi*⁺ cell markers were already specified in the text as *histone h3* (PrileiEvm022498t1) and *piwi* (PrileiEvm003567t1). We have included these ortholog names in Figure 7.

(3) Finally, regarding the expression of marker PrileiEvm025662t1, we consider that the pattern observed in the fission zone is perfectly valid, as this area generates all the cell types of the new individual. The fission zone and the posterior growth zone are also more prone to show colocalization between *piwi*⁺ cells and other cell types, as these are highly proliferative areas with more differentiation events. In the image shown, the right side of the fission zone is generating the new head, and therefore has more neurons than the left side, which is giving rise to the new tail instead. However, the signal observed from PrileiEvm025662t1 could correspond to both neurons and polycystin cells.

- Is it possible that there are lineage-committed stem cell populations that are *piwi*⁻ (e.g. epidermal) that do not come from a *piwi*⁺ stem cell pool? The authors kind of suggest it in the discussion, but it is not really in Figure 8.

We have changed Figure 8 to include this possibility. To clarify this point, we have reformulated the last sentence referring to Figure 8C:

'However, further work is needed to determine if these epidermal cells are piwi+, if they are a stem cell population capable of self-renewal and if they constitute a niche isolated from the main piwi+ stem cell pool.'

- Line 85: "we characterised the molecular signature of annelid *piwi*⁺ cells at both the transcriptional and epigenetic level". The authors do not technically characterise *piwi*⁺ cells at the epigenetic level. They assess the expression of epigenetic regulators in *piwi*⁺ cells, but we do not know how these cells differ epigenetically from the rest of differentiated cell types. I suggest rewording as "we characterised the molecular signature of annelid *piwi*⁺ cells at the transcriptional level"

Changed to: Finally, we characterised the molecular signature of annelid *piwi*⁺ cells at the transcriptional level, revealing a transcriptional program composed of RNA binding proteins, cell cycle control, DNA repair mechanisms, and chromatin regulators.

- Line 160: "involved" instead of "including"?

Changed

- Lines 232-233: "and showcases the power of ACME and SPLiT-seq to unravel the TF expression underlying cell type differentiation or function in uncharacterised species." This is an odd sentence, especially when the authors admit in the first sentence of that paragraph that their combinatorial single-cell dataset had a low UMI and gene count, and thus, they decided to use a pseudobulk approach to investigate transcription factor expression. I suggest deleting the phrase in lines 232-233.

Thank you for the suggestion. The sentence has been removed.

- Line 260: change "reinforcing the notion" to "reinforcing the scenario"

Changed

- Line 372: "piwi" in italics at the end of the line.

Changed

Reviewer #1 (Remarks on code availability):

The repository seems to be complete and well explained

REVIEWER 2

Reviewer #2 (Remarks to the Author):

Authors have revised and implemented all of the points that were raised carefully— they have expanded their hypothesis framework, spearheaded by the addition of the new discussion figure (Figure 8), which highlights the diverse hypotheses that are emergent from their findings. In addition, authors performed EdU experiments that explore the cellular properties, including their proliferative identity, proportion and distribution in the *Pristina* body plan. Finally, authors understood the issues raised with language in the manuscript, mainly with the “reconstruction of the differentiation trajectories” using PAGA.

While these revisions certainly strengthen the manuscript, it is important that authors detail all possible hypotheses in the text; the presence of the discussion figure helps with the conceptualization of the hypotheses that emerge from the study, but it is important to not favor any hypothesis over any other one in the text. For instance, the sentence “Altogether, these analyses showed that *piwi*⁺ cells in *Pristina* are a heterogeneous cell population with a pluripotent stem cell signature that is also observed in other pluripotent stem cells.” is presented as one that only considers the hypothesis conceptualized in Figure 8A, but does not consider hypotheses 8B, 8C, or 8D.

This sentence is presented after the analysis that show that a) several differentiation lineages can be reconstructed from *piwi*⁺ cells, b) this leads to a high potency score, c) several subpopulations of *piwi*⁺ cells can be detected,

expressing a mixture of stem and differentiation markers. As the referee pointed out these observations do not show their pluripotency, but they largely parallel the observations of Plass et al and Fincher et al in 2018 on planarian stem cells, which had previously been shown to contain pluripotent cells by Wagner et al in 2011. Therefore, we feel that these observations do favour the 8A and would like to point out that in the text. We have rephrased the sentence that the referee pointed out to more clearly phrase what the data shows and put it into the right context:

“Altogether, these analyses showed that piwi+ cells in Pristina are a heterogeneous cell population with transcriptomic properties that are also observed in other pluripotent stem cells. However, individual cell potencies need to be demonstrated in future studies”

Balancing the emergent hypotheses presented in this paper will allow for room to find commonalities in other annelid studies, such as the Stockinger et al manuscript that will be co-published with this article— stressing one hypothesis over any other ones will limit these commonalities. Although these papers focus on different species, and it is entirely possible that different cellular dynamics operate across annelids, given that the data in this manuscript cannot rule out multiple hypotheses, it is important to not leave the reader with just one major interpretation. For example, given that functional perturbation of the piwi+ population has not been shown, it is a stretch to definitely state that the piwi+ cells “underlie” differentiation, as stated in the abstract. The last sentence in the abstract could be reworded to read “Our data reveal the cell type diversity of adult annelids by single cell transcriptomics and suggest that a piwi+ cell population with a pluripotent stem cell signature is associated with adult cell type differentiation.”

The last sentence of the abstract has been changed as suggested.

REVIEWER 3

Reviewer #3 (Remarks to the Author):

In general, the authors have engaged with our suggestions, which is very positive. Some open questions/points remain however.

For the TF analysis in Fig.4, the authors did not address my concern, as the TFs best representing each cluster can still not be identified. Please add a plot with names at least to the Supplementary material.

We have addressed this concern by modifying Supplementary File 9. This file contains a list of all TFs, and for each TF we give annotations (with orthology, PFAM domains, TF Class, etc). We flag those TFs in Figure 4C in a column, and for each of these we give the cell cluster with highest expression, and the associated expression data. This table contains a variable number of TFs for each cluster, but we have now added two new columns: 1) a column flagging the TFs that are top central in each cluster, following the analysis of Figure 4G (please note that not all clusters have top central TFs) 2) a column flagging the TF of each cluster that has the highest fold change. Researchers would then be able to use these two sources of information (as well as work with the numbers originally provided in Supplementary File 9).

Also, I am not sure about the insistence on the UMAPs: for example, the dotblots shown as text figures cannot convey the message because “gut” is only featured by one column. Do the TFs by any means relate to the AP sections of the gut? I insist that while the dot plots might overemphasize cell types with relatively few cells, at least we are getting some true data regarding specificity for a given type, for which it does not matter how frequent that type is, and that is the purpose of the figure. UMAPs are usually not chosen because colored dots are often plotted on top of each other and are hence invisible depending on the population density of that cluster within expression space - you are trying to compress 80k points in 5cm x 5cm pictures.

For PrileiEVm000093t1, a zinc finger, the UMAP suggests some expression in the piwi+ cells and in the general epidermis part, with very few random occurrences of expression elsewhere; the dotplot however suggests significant expression e.g. in 10-20% of lumbrokinase+ cells or Idrr+ cells. In general, the dotplots illustrate much better that the piwi+ cells share significant expression of multiple TFs with different clusters. I would thus urge usage of dot plots; alternatively, the authors may consider domino plots.

We understand the reviewer’s concern, and would like to say that when such highly dimensional datasets are used, very few representations are flawless. Oftentimes some are better than others in some respects, but underperform in others. For instance, in the case highlighted by the reviewer (PrileiEVm000093t1)

the dotplot misrepresents the data because of two different aspects: 1) The size of the dot represents the fraction, and the colour represents the mean expression. However, for sizes of circle that are very small, the colour is very difficult to judge. The sizes of the dots in that particular example might be similar (indicating similar proportions) but the mean expression might be smaller, which can only be seen in the grey to blue relative level. Indeed, the expression in lumbrokinase is 0.044, compared to 0.067 in piwi+ cells, and dwarfed by the larger 0.1623 in epidermis. This results in two very similar shades of blue-grey, that are packed in a small dot, but the expression in the lumbrokinase is 30% lower than in piwi+ cells. 2) As the reviewer correctly acknowledges, the size of the dot is the fraction, and it is problematic when comparing larger clusters to smaller ones. The TF in question is expressed in a percentage of cells similar to the percentage in piwi+ cells, but the numbers are affected by the small size of the lumbrokinase+ cell cluster: the TF is expressed in only 2 cells out of 603, a 0.33%, which is similar to the 0.40% value in piwi+ cells, but this number comes from a 66 out of 16,247 ratio. The value in the lumbrokinase+ cells is therefore affected by the very small count of 2, and it would be highly affected by receiving one count more or less. This information is not accessible in the dotplot and the referee or the readers would not have been able to check. The UMAPs, for these shallow numbers, display the information better. Upon inspection one can see that there are only 2 cells expressing the factor in the lumbrokinase+ cells (looking at Figure 4D). Our plotting engine plots positive values (“blue” dots) on top of zero values (“grey” dots) (by using the default “sort order” parameter of scanpy, which “*for continuous annotations used as color parameter, plot data points with higher values on top of others*”), and therefore the UMAP gives a better impression of the raw data in this precise example. We agree that in a more crowded UMAP plot these dots would overlap as the referee points out and give a false impression. After all, we present both the UMAPs and the dotplots, which are included in the response to the reviewers and will be published if the paper is accepted.

The authors did quite some work on the former Fig. 4E, clarifying that the apparent connectivity between stomach-neurons-secretory-intestine was an artifact indeed. This is a significant improvement.

We thank the reviewer for this comment, as we were prompted by their insightful comment to look further into the data and we are glad that, with the reviewers help, we managed to improve the manuscript.

I would encourage the authors to include at least some of the extra work in some form, either as supplementary notes or as notebooks on their GitHub repository (which is finally public).

We appreciate the reviewer’s acknowledgement of these analyses. We are currently working on a separate project that explores these questions more in depth and in different data. We will update the GitHub repository upon acceptance. We continue our improvement path and will present a newer version of the code for the WGCNA analysis in this new manuscript.

It remains unfortunate that the most interesting point, the heterogeneity of the piwi+ cells, does not emerge from the data more clearly, and that the double HCRs did not work so well.